# CoSTA*: Cost-Sensitive Toolpath Agent for Multi-turn Image Editing

## Abstract

Text-to-image models like Stable Diffusion and DALLE-3 still struggle with complex multi-turn image editing. We study how to break down such a task into a sequence of subtasks and address them by an agentic workflow (path) of AI tool use with minimum costs. Conventional search algorithms require expensive exploration to find tool paths. While large language models (LLMs) possess prior knowledge of subtask planning, their estimation of the quality and cost of tools is usually inaccurate to determine which to apply in each subtask. *Can we combine the strengths of both LLMs and graph search to find cost-efficient tool paths?* We propose a three-stage approach "CoSTA*" that leverages LLMs to create a subtask tree that prunes a graph of AI tools for the given task, and then conducts A* search on the small subgraph to find a tool path. To better balance the total cost and quality, CoSTA* combines both metrics of each tool on every subtask to guide the A* search. Each subtask's output is evaluated by a vision-language model (VLM), where a failure will trigger an update of the tool's cost and quality on that subtask. Hence, the A* search can recover from failures quickly to explore other paths. Moreover, CoSTA* can automatically switch between modalities across subtasks for a better cost-quality trade-off. We build a novel benchmark of challenging multi-turn image editing, on which CoSTA* outperforms state-of-the-art image-editing models or agents in both cost and quality, and performs versatile trade-offs upon user preference. Our dataset and a hosted demo can be found here.

## 1 Introduction

Text-to-Image models such as stable diffusion, FLUX, and DALLE (Ramesh et al., 2021) has been widely studied to replace humans on image-editing tasks, which are time-consuming due to various repetitive operations and trial-and-errors. While these models have exhibited remarkable potential for generating diverse images and simple object editing, they usually struggle to follow composite instructions that require multi-turn editing, in which a sequence of delicate adjustments are requested to manipulate (e.g., remove, replace, add) several details (e.g., object attributes or texts) while keeping other parts intact. For example, given an image, it is usually challenging for them to "recolor the chalkboard to red while redacting the text on it and write "A CLASSROOM" on the top. Also, detect if any children are in the image."

Although a large language model (LLM) can decompose the above multi-turn composite task into easier subtasks, and each subtask can be potentially learned by existing techniques such as ControlNet, the required training data and computational costs are usually expensive. Hence, a training-free agent that automatically selects tools to address the subtasks is usually more appealing. However, finding an efficient and successful path of tool use (i.e., toolpath) is nontrivial, and as our experiments demonstrate, current agents often fail to plan efficiently for complex, high-turn editing tasks. While some subtasks are exceptionally challenging and may require multi-round trial-and-error with advanced and costly AI models, various subtasks could be handled by much simpler, lower-cost tools. Moreover, users with limited budgets usually prefer to control and optimize the trade-off between quality and cost. However, most existing image-editing agents are not cost-sensitive, so the search cost of their toolpaths can be highly expensive.

Despite the strong heuristic of LLMs on tool selection for each subtask, as shown in Figure 2, they also suffer from hallucinations and may generate sub-optimal paths due to the lack of precise

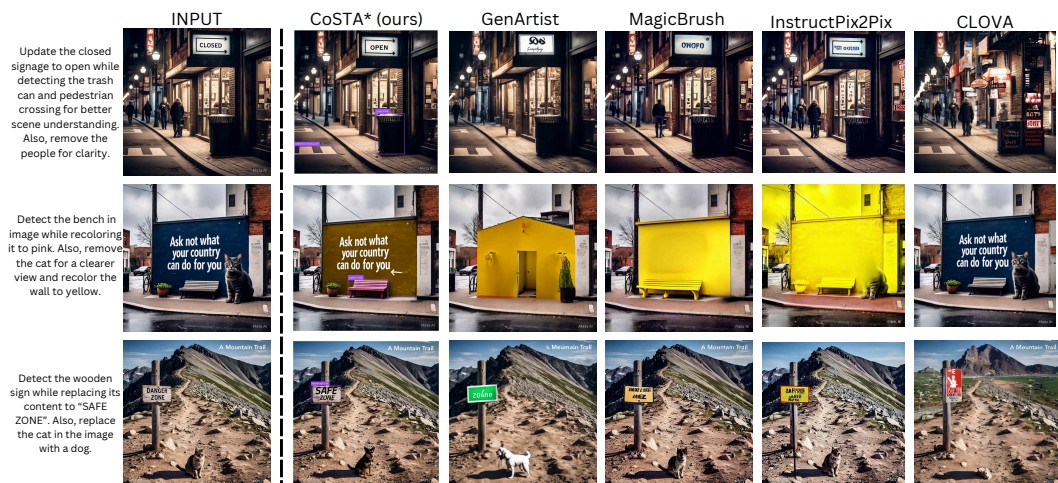

Figure 1: Comparison of CoSTA* with State-of-the-Art image editing models/agents, which include GenArtist (Wang et al., 2024b), MagicBrush (Zhang et al., 2024a), InstructPix2Pix (Brooks et al., 2023), and CLOVA (Gao et al., 2024). The input images and prompts are shown on the left of the figure. The outputs generated by each method illustrate differences in accuracy, visual coherence, and the ability to multimodal tasks. Figure 8 shows examples of step-by-step editing using CoSTA*with intermediate subtask outputs presented. Some extra comparisons with the recent Gemini 2.0 Flash can be seen in Figure 11.

knowledge for each tool and the long horizon of multi-turn editing. On the other hand, classical search algorithms such as A* and MCTS can precisely find the optimal tool path after sufficient exploration, if accurate estimates of per-step value/cost and high-quality heuristics are available. However, they are not scalable to explore tool paths on a large-scale graph of many computationally heavy models as tools, e.g., diffusion models. This motivates the question: *Can we combine the strengths of both methods in a complementary manner?*

In this paper, we develop a novel agentic mechanism "**Cost-Sensitive Toolpath Agent (CoSTA*)**" that combines both LLMs and A* search's strengths while overcoming each other's weaknesses to find a cost-sensitive path of tool use for a given task. As illustrated in Figure 2, we propose a hierarchical planning strategy where an LLM focuses on subtask planning (each subtask is a subsequence of tool uses), which decomposes the given task into a subtask tree on which every root-to-leaf path is a feasible high-level plan for the task. This is motivated by the observations that LLMs are more powerful on subtask-level commonsense reasoning but may lack accurate knowledge to decide which specific tools to use per subtask. Then, a

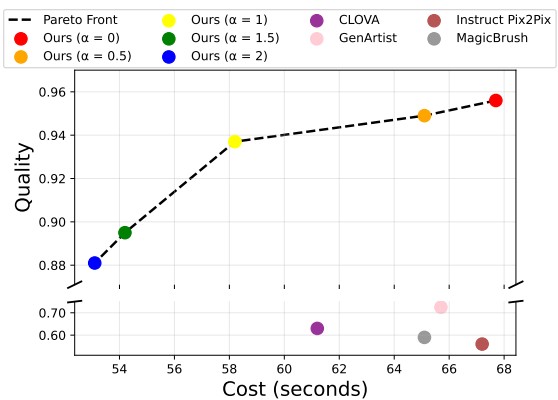

Figure 3: CoSTA* with different cost-quality trade-off coefficients $\alpha$ vs. four recent image-editing models/agents. CoSTA* achieves Pareto optimality and dominates baselines on both metrics.

low-level A* search is applied to the subgraph spanned by the subtask tree on a tool dependency graph (TDG, with an example in Figure 4). It aims to find a toolpath fulfilling the user-defined quality-cost trade-off. The subtask tree effectively reduces the graph of tools on which the A* search is conducted, saving a significant amount of searching cost.

In CoSTA*, we exploit available prior knowledge and benchmark evaluation results of tools, which are underexplored in previous LLM agents, to improve both the planning and search accuracy. We mainly leverage two types of prior information: (1) the input, output, and subtasks of each tool/model; and (2) the benchmark performance and cost of each tool or model reported in the existing literature.

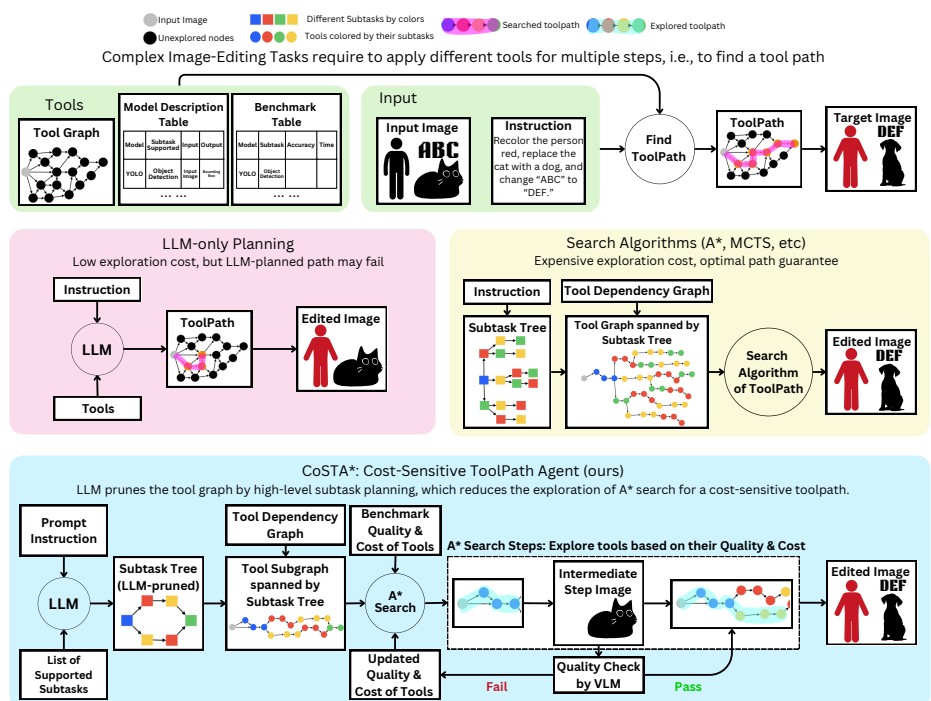

Figure 2: Comparison of CoSTA* with other planning agents. LLM-only planning is efficient but prone to failure and heuristics. Search algorithms like A* guarantee optimal paths but are computationally expensive. CoSTA* balances cost and quality by first pruning the subtask tree using an LLM, which reduces the graph of tools we conduct fine-grained A* search on.

Specifically, a sparse tool dependency graph (TDG) is built based on (1), where two tools are connected if the first's output is a legal input to the second in certain subtask(s). Moreover, the information in (2) defines the heuristics $h(x)$ in A* search, which combines both the cost and quality with a trade-off coefficient $\alpha$. We further propose an actual execution cost $g(x)$ combining the actual cost and quality in completed subtasks, and update it during exploration. By adjusting $\alpha$, the cost-sensitive A* search aims to find a toolpath aligning with user preference of quality-cost trade-off.

To examine the performance of CoSTA*, we curate a novel benchmark for multi-turn image editing with challenging, composite tasks. We compare CoSTA* with state-of-the-art image-editing models or agents. As shown in Figure 3, CoSTA*achieves advantages over others on both the cost and quality, pushing the Pareto frontier of their trade-offs. In Figure 8, in several challenging multi-turn image-editing tasks, only CoSTA* accomplishes the goals. **Our main contributions and novelties can be summarized as below** (More detailed list of novelties and contributions can be found in Appendix B with detailed motivations in Appendix Q.):

- We propose a novel hierarchical planning agent CoSTA* that combines the strengths of LLMs and graph search to find toolpaths for composite multi-turn image editing.
- CoSTA* addresses the quality-cost trade-off problem by a controllable cost-sensitive A* search and employing a novel definition of the cost-quality formulation, and achieves the Pareto optimality over existing agents.
- We are able to achieve great results on text-in-image editing tasks by supporting multimodality.
- We exploit prior knowledge of tools to improve the toolpath finding.
- We propose a new challenging benchmark for multi-turn image editing covering tasks of different complexities.

## 2 RELATED WORK

**Image Editing via Generative AI** Image editing has seen significant advancements with the rise of diffusion models (Dhariwal & Nichol, 2021; Ho et al., 2020), enabling highly realistic and diverse image generation and modification. Modern approaches focus on text-to-image frameworks

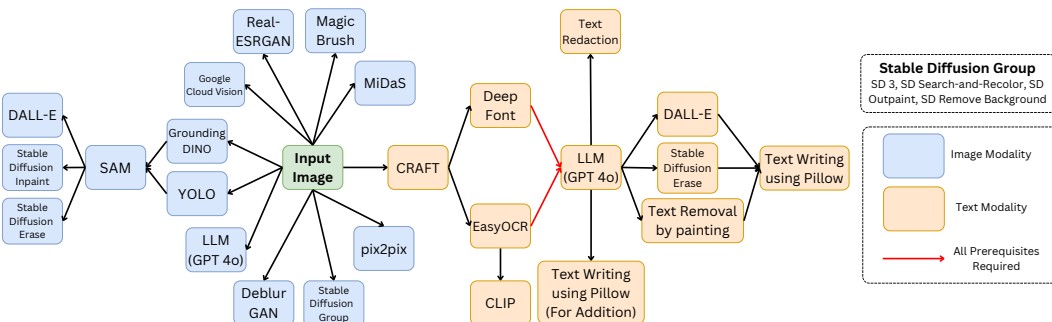

Figure 4: **Tool Dependency Graph (TDG)**. A directed graph where nodes represent tools and edges indicate dependencies. An edge $(v_1, v_2)$ means $v_1$'s output is a legal input of $v_2$. It enables toolpath search for multi-turn image-editing tasks with composite instructions.

that transform descriptive text prompts into images, achieving notable quality (Chen et al., 2023a; Rombach et al., 2022b; Saharia et al., 2022) but often facing challenges with precise control over outputs. To mitigate this controllability issue, methods like ControlNet (Zhang et al., 2023b) and sketch-based conditioning (Voynov et al., 2022) refine user-driven edits, while layout-to-image systems synthesize compositions from spatial object arrangements (Chen et al., 2023b; Li et al., 2023b; Lian et al., 2024; Xie et al., 2023). Beyond text-driven editing, research efforts have also focused on personalized generation and domain-specific fine-tuning for tasks such as custom content creation or rendering text within images. However, current models still struggle with handling complex prompts, underscoring the need for unified, flexible solutions (Brooks et al., 2023; Chen et al., 2024; Parmar et al., 2023; Yang et al., 2022).

**Large Multimodal Agents for Image Editing** Recent advancements in multimodal large language models (MLLMs) have significantly enhanced complex image editing capabilities (Wang et al., 2024b; Huang et al., 2024; Zhang et al., 2024c; Huang et al., 2023; Zhang et al., 2024d; Yang et al., 2024; Wang et al., 2024c). GenArtist (Wang et al., 2024b) introduces a unified system where an MLLM agent coordinates various models to decompose intricate tasks into manageable sub-problems, enabling systematic planning and self-correction. DialogGen (Huang et al., 2024) aligns MLLMs with text-to-image (T2I) models, facilitating multi-turn dialogues that allow users to iteratively refine images through natural language instructions. IterComp (Zhang et al., 2024c) aggregates preferences from multiple models and employs iterative feedback learning to enhance compositional generation, particularly in attribute binding and spatial relationships. SmartEdit (Huang et al., 2023) leverages MLLMs for complex instruction-based editing, utilizing a bidirectional interaction module to improve understanding and reasoning. These approaches build upon foundational works like BLIP-2 (Li et al., 2023a), which integrates vision and language models for image understanding, and InstructPix2Pix (Brooks et al., 2023), which focuses on text-guided image editing.

## 3 FOUNDATIONS OF CoSTA*

We present the underlying models, supporting data structures, and prior knowledge that CoSTA* relies on before explaining the design of the CoSTA* algorithm. Specifically, we describe the Model Description Table, the Tool Dependency Graph, and the Benchmark Table.

### 3.1 MODEL DESCRIPTION TABLE

Table 1: Model Description Table (excerpt)

| Model | Supported Subtasks | Inputs | Outputs |
|---|---|---|---|
| YOLO (Wang et al., 2022) | Object Detection | Input Image | Bounding Boxes |
| SAM (Kirillov et al., 2023a) | Segmentation | Bounding Boxes | Segmentation Masks |
| DALL-E (Ramesh et al., 2021) | Object Replacement | Segmentation Mask | Edited Image |
| Stable Diffusion Inpaint (Rombach et al., 2022a) | Object Removal, Replacement, Recoloration | Segmentation Mask | Edited Image |
| EasyOCR (Kittinaradorn et al., 2022) | Text Extraction | Text Bounding Box | Extracted Text |

We first construct a Model Description Table (MDT) that lists all specialized models (e.g., SAM, YOLO) and the corresponding tasks they support (e.g., image segmentation, object detection). In this paper, we consider 24 models that collectively support 24 tasks, covering both image and text modalities. The supported tasks can be broadly categorized into *image editing* tasks (e.g., object removal, object recolorization) and *text-in-image editing* tasks (e.g., text removal, text replacement). Our system allows for easy extension by adding new models and their corresponding tasks to this table. The MDT also includes columns specifying the input dependencies and outputs of each model. An excerpt of the MDT is shown in Table 1 to illustrate its structure, and full MDT is available in Appendix (Table 18).

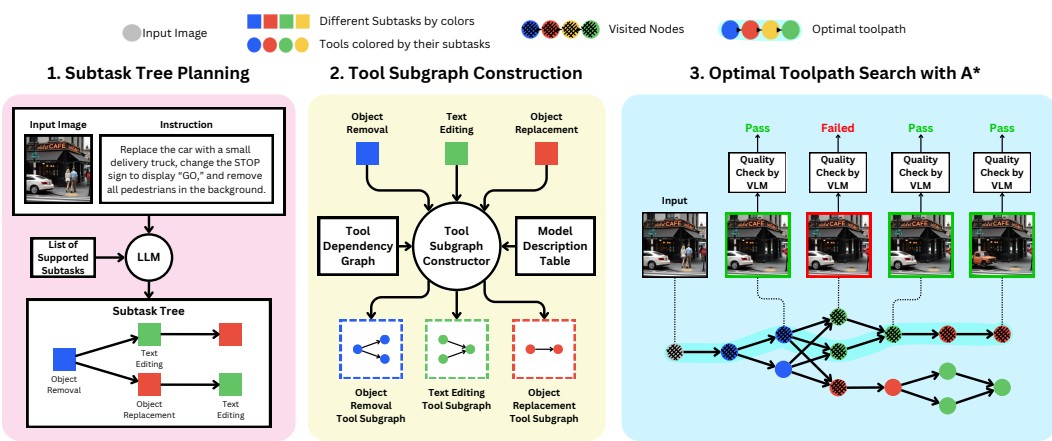

Figure 5: **Three stages in CoSTA**$^*$: (1) an LLM generates a subtask tree based on the input and task dependencies; (2) the subtask tree spans a tool subgraph that maintains tool dependencies; and (3) A$^*$ search finds the best toolpath balancing efficiency and quality.

### 3.2 TOOL DEPENDENCY GRAPH

Each tool in our library is a specialized model for a specific subtask, where some tools require the outputs of other tools as inputs. To capture these dependencies, we construct a Tool Dependency Graph (TDG). Formally, we define the TDG as a directed graph $G_{td} = (V_{td}, E_{td})$, where $V_{td}$ is the set of tools, and $E_{td} \subseteq V_{td} \times V_{td}$ contains edges $(v_1, v_2)$ if tool $v_2$ depends on the output of $v_1$. Figure 4 presents the full TDG, illustrating the dependencies between tools. This TDG can be automatically generated based on the input-output specifications of each tool mentioned in the MDT, reducing the need for extensive human effort (see Appendix G for a detailed explanation).

### 3.3 BENCHMARK TABLE FOR HEURISTIC SCORES

At its core, CoSTA$^*$ employs A$^*$ search over a network of interdependent tools to find the optimal cost-sensitive path. This process relies on a heuristic function $h(x)$ for each tool $x$. We initialize these heuristic values using prior knowledge of execution time and quality scores obtained from existing benchmarks or published studies (e.g., mAP score for YOLO (Wang et al., 2022)). Since not all tools have sufficient benchmark data, we evaluate them over **137 instances of the specific subtask**, applied across **121 images from the dataset** to handle missing values. These initial heuristics can be derived either from such offline experiments or dynamically via a "cold start" approach where the table is populated by aggregating real-time feedback ($g(x)$) from inference (Appendix R). For each tool-task pair $(v_i, s_j)$, we define an execution time $C(v_i, s_j)$ and a quality score $Q(v_i, s_j)$. To ensure comparability, quality values are normalized per subtask to a $[0, 1]$ scale (See Appendix T for rationale). The complete Benchmark Table (BT) is shown in Table 19.

## 4 COSTA$^*$: COST-SENSITIVE TOOLPATH AGENT

This section details our approach for constructing and optimizing a Tool Subgraph (TS) to efficiently execute multimodal editing tasks. The methodology consists of three key stages: (1) generating a subtask tree, (2) constructing the TS, and (3) applying A$^*$ search to determine the optimal execution path.

First, as shown in Figure 5, an LLM infers subtasks and dependencies from the input image, prompt, and the set of supported subtasks $\mathcal{S}$, generating a subtask tree $G_{ss}$. Then, this tree is transformed into the Tool Subgraph $G_{ts}$, where each subtask is mapped to a model subgraph within the TDG. This ensures that model dependencies are maintained while incorporating task sequences and execution constraints. Finally, A$^*$ search explores $G_{ts}$ to identify an optimal execution path by balancing computational cost and output quality. It prioritizes paths based on a cost function $f(x) = g(x) + h(x)$ where $g(x)$ represents real-time execution costs, and $h(x)$ is the precomputed heuristic. A tunable parameter $\alpha$ controls the tradeoff between efficiency and quality, allowing for adaptive optimization.

### 4.1 TASK DECOMPOSITION & SUBTASK TREE PLANNING

Given an input image $x$ and prompt $u$, we employ an LLM $\pi(\cdot | f_{plan}(x, u, \mathcal{S}))$ to generate a subtask tree $G_{ss} = (V_{ss}, E_{ss})$, where each node $v_i$ represents a subtask $s_i$, and each edge $(v_i, v_j)$ denotes a dependency. Here, $f_{plan}$ is a prompt template containing the input image, task description $u$, and

supported subtasks $\mathcal{S}$. The full prompt is detailed in Appendix X. The LLM infers task relationships, forming a directed acyclic graph where each root-to-leaf path represents a valid solution.

The subtask tree encodes various solution approaches, accommodating different subtask orders and workflows. Path selection determines an optimized workflow based on efficiency or quality. Part 1 of Figure 5 (Subtask Tree Planning) illustrates an example where the LLM constructs a subtask tree from an input image and prompt.

### 4.2 Tool Subgraph Construction

The TS, denoted as $G_{ts} = (V_{ts}, E_{ts})$, represents the structured execution paths for fulfilling subtasks in the *Subtask Tree* (ST) $G_{ss}$. It is constructed by mapping each subtask node to a corresponding model subgraph from the TDG $G_{td}$.

The *node set* $V_{ts}$ consists of all models required for execution, ensuring that every subtask $s_i \in S$ is associated with a valid model:

$$V_{ts} = \bigcup_{s_i \in S} M(s_i), \tag{1}$$

where $M(s_i)$ denotes the set of models that can perform subtask $s_i$, as listed in the MDT.

The *edge set* $E_{ts}$ represents dependencies between models, ensuring that each model receives the necessary inputs from its predecessors before execution. These dependencies are derived from $G_{td}$ by backtracking to identify required intermediate outputs:

$$E_{ts} = \bigcup_{s_i \in S} E_{ti}, \tag{2}$$

where $E_{ti}$ contains directed edges between models in $M(s_i)$ based on their execution dependencies. The final tool subgraph $G_{ts}$ encapsulates all feasible execution paths while preserving dependencies and logical consistency. Figure 5 (Tool Subgraph Construction) illustrates this transformation.

### 4.3 Path Optimization with A* Search

The A* algorithm finds the optimal execution path by minimizing the cost function: $f(x) = g(x) + h(x)$ where $g(x)$ is the **actual execution cost**, dynamically updated during execution, and $h(x)$ is the **heuristic estimate**, precomputed from benchmark values. Nodes are explored in increasing order of $f(x)$, ensuring an efficient tradeoff between execution time and quality.

### 4.4 Heuristic Cost $h(x)$

The heuristic cost $h(x)$ estimates the best-case execution cost from node $x$ to a leaf node (excluding the cost of $x$ itself), factoring in both execution time and quality. Each node represents a tool-task pair $(v_i, s_i)$, where $v_i$ is the tool and $s_i$ is the subtask. For example, $y = (\text{YOLO}, \text{Object Detection})$ ensures that $y$ is inherently multivariate. The heuristic is defined as:

$$h(x) = \min_{y \in \text{Neighbors}(x)} [h_C(y) + C(y)]^\alpha \times [2 - Q(y) \times h_Q(y)]^{(2-\alpha)} \tag{3}$$

where $h_C(y)$ represents the cost component of $h(y)$ (initialized as 0 for leaf nodes), while $h_Q(y)$ denotes the quality component (initialized as 1 for leaf nodes). $C(y)$ and $Q(y)$ correspond to the benchmark execution time and quality of tool $y$, respectively, and $\alpha$ controls the tradeoff between cost and quality. This heuristic propagates recursively, ensuring each node maintains the best possible estimate to a leaf node.

### 4.5 Actual Execution Cost $g(x)$

The actual execution cost $g(x)$ is computed in real-time as execution progresses:

$$g(x) = \left( \sum_{i=1}^{x} c(v_i, s_i) \right)^\alpha \times \left( 2 - \prod_{i=1}^{x} q(v_i, s_i) \right)^{2-\alpha} \tag{4}$$

where $c(v_i, s_i)$ represents the actual execution time (in seconds) of the tool-subtask pair $(v_i, s_i)$, and $q(v_i, s_i)$ is the real-time validated quality score for the same pair.

The summation includes only nodes in the currently explored path. Each node is initialized with $g(x) = \infty$, except the start node, which is set to zero. Upon execution, $g(x)$ is updated to the minimum observed value.

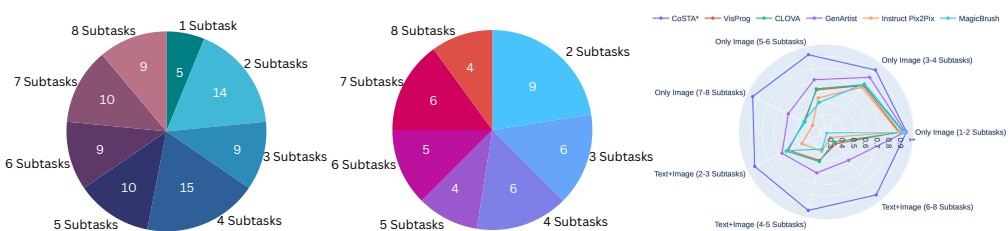

Figure 6: Distribution of image-only (left) and text+image tasks (middle) in **our proposed benchmark, and quality comparison** of different methods on the benchmark (right). CoSTA* excels in complex multimodal tasks and outperforms all the baselines.

If a node $x$ fails the manually set quality threshold, it undergoes a retry mechanism with updated hyperparameters. If successful, the new execution cost is accumulated in $g(x)$. If a node fails all retries, $g(x)$ remains unchanged, and the path is not added back to the queue, ensuring failed paths are deprioritized, but alternative routes exploring the same node remain possible. More information about the execution is in Appendix O along with detailed justification for the formulation of equations 3 & 4 in Appendix P.

## 5 EXPERIMENTS

We evaluate CoSTA* on a curated dataset and the MagicBrush benchmark (Zhang et al., 2024a), comparing it against baselines to assess its effectiveness in complex image and text-in-image editing. (All experiments have been conducted on a single NVIDIA A100 GPU)

### 5.1 EXPERIMENTAL SETTINGS

**Benchmark Dataset** Our dataset consists of 121 manually curated images with prompts involving 1–8 subtasks per task (**amounting to 550 total image manipulations or turns**), ensuring comprehensive coverage across both image and text-in-image modalities. It includes 81 tasks with image-only edits and 40 tasks requiring multimodal processing. Figure 6 summarizes its even distribution, with further details in Appendix H. We also evaluate CoSTA* on the **MagicBrush** (Zhang et al., 2024a) and **EMU-Edit** (Sheynin et al., 2023) benchmarks for both single and multi-turn tasks (Appendix A).

**Baselines** We compare CoSTA* against agentic baselines such as VISPROG (Gupta & Kembhavi, 2023), GenArtist (Wang et al., 2024a), and CLOVA (Gao et al., 2024). These methods support task orchestration but lack CoSTA*'s A* path optimization, cost-quality tradeoff, and multimodal capabilities. For InstructPix2Pix (Brooks et al., 2023) and MagicBrush (Zhang et al., 2024a), not inherently designed for single-pass multi-turn instructions, we applied them iteratively for multi-step edits, which could increase their execution time relative to specialized multi-step agents. We also compare CoSTA* with latest closed-source models like Gemini 2.0 Flash (Gemini, 2024) and GPT-4o (OpenAI, 2024) with detailed quantitative results in Appendix I.

### 5.2 EVALUATION METRICS

Table 2: Comparison of CLIP Similarity vs. Human Evaluation on 50 tasks.

| Metric | CLIP Score | Human Acc. |
|---|---|---|
| Avg. (50 Tasks) | 0.96 | 0.78 |

**Human Evaluation** To ensure a reliable assessment of model performance, we employ human evaluation for accuracy measurement. Each subtask $s_i$ in task $T$ is manually assessed and assigned a score $A(s_i)$: 1 if fully correct, 0 if failed, and $x \in (0, 1)$ if partially correct. Task-level accuracy $A(T)$ is computed as the mean of its subtasks, while overall accuracy $A_{\text{overall}}$ is averaged over all evaluated tasks. For partial correctness ($x$), predefined rules are used to assign values based on specific evaluation criteria. This structured human evaluation provides a robust performance measure across all tasks (see Appendix F for a detailed explanation of the evaluation process and the rules for assigning partial scores).

**Human Evaluation vs. CLIP Scores** While automatic metrics like CLIP similarity are common for image/text editing, we use human evaluation for complex, multi-step, multimodal tasks. CLIP often misses small but critical changes (e.g., missing bounding boxes) and struggles with semantic coherence in multimodal tasks or tasks with multiple valid outputs. Our evaluation of 50 tasks with

Table 3: Accuracy comparison of CoSTA* with baselines across task types and categories. CoSTA* excels in complex workflows with A* search and a diverse set of tools. (All values are at $\alpha = 1$.)

| Task Type | Task Category | CoSTA* | VisProg | CLOVA | GenArtist | Instruct Pix2Pix | MagicBrush |
|---|---|---|---|---|---|---|---|
| **Image-Only Tasks** | 1–2 subtasks | **0.94** | 0.88 | 0.91 | 0.93 | 0.87 | 0.92 |
| | 3–4 subtasks | **0.93** | 0.76 | 0.77 | 0.85 | 0.74 | 0.78 |
| | 5–6 subtasks | **0.93** | 0.62 | 0.63 | 0.71 | 0.55 | 0.51 |
| | 7–8 subtasks | **0.95** | 0.46 | 0.45 | 0.61 | 0.38 | 0.46 |
| **Text+Image Tasks** | 2–3 subtasks | **0.93** | 0.61 | 0.63 | 0.67 | 0.48 | 0.62 |
| | 4–5 subtasks | **0.94** | 0.50 | 0.51 | 0.61 | 0.42 | 0.40 |
| | 6–8 subtasks | **0.94** | 0.38 | 0.36 | 0.56 | 0.31 | 0.26 |
| **Overall Accuracy** | Image Tasks | **0.94** | 0.69 | 0.70 | 0.78 | 0.64 | 0.67 |
| | Text+Image Tasks | **0.93** | 0.49 | 0.50 | 0.61 | 0.40 | 0.43 |
| | All Tasks | **0.94** | 0.62 | 0.63 | 0.73 | 0.56 | 0.59 |

intentional errors showed CLIP similarity scores (0.93-0.98) significantly higher than human accuracy (0.7-0.8), highlighting CLIP's limitations (Table 2).

**CLIP in Feedback Loops vs. Dataset Evaluation** CLIP is effective for real-time subtask validation, as each subtask is assessed in isolation. In object detection, for instance, it evaluates only the detected region against the expected label (e.g., 'car' or 'person'), ensuring accurate verification. However, for full-task evaluation, CLIP prioritizes global similarity, often missing localized errors, making it unreliable for holistic assessment but useful for individual subtasks.

Table 4: Correlation Analysis of CLIP vs Human Evaluation on 40 tasks, which indicates that human evaluation is still necessary.

| Metric | Correlation Coefficient | $p$-value |
|---|---|---|
| Spearman's $\rho$ | 0.59 | $6.07 \times 10^{-5}$ |
| Kendall's $\tau$ | 0.47 | $5.83 \times 10^{-5}$ |

**Correlation Analysis** We analyzed the correlation between CLIP scores and human accuracy across 40 tasks, finding weak agreement (Spearman's $\rho = 0.59$, Kendall's $\tau = 0.47$). The low correlation confirms CLIP's inability to capture nuanced inaccuracies, as visualized in Table 4 and the scatter plot in Appendix N.

**Execution Cost (Time)** The cumulative execution time, including feedback-based retries and exploration of alternate models, is used to evaluate CoSTA*'s efficiency.

### 5.3 MAIN RESULTS

Table 3 demonstrates that CoSTA* consistently outperforms baselines across all task categories. For simpler image-only tasks (1–2 subtasks), CoSTA* achieves comparable accuracy, but as complexity increases (5+ subtasks), it significantly outperforms baselines. This is due to its A* search integration, which effectively refines LLM-generated plans, whereas baselines struggle with intricate workflows.

In text+image tasks, CoSTA* achieves much higher accuracy due to its extensive toolset for text manipulation. Baselines, limited in tool variety, fail to perform well in multimodal scenarios. Additionally, CoSTA*'s dynamic feedback and retry mechanisms further enhance robustness across diverse tasks, maintaining high-quality outputs. These results highlight its superiority in balancing cost and quality over agentic and non-agentic baselines.

Table 5: Comparison of key features across methods, highlighting the capabilities supported by CoSTA*, which are absent in baselines and contribute to its superior performance.

| Feature | CoSTA* | CLOVA | GenArtist | VisProg | Instruct Pix2Pix |
|---|---|---|---|---|---|
| System Architecture | Agent Based | Agent Based | Agent Based | Agent Based | End-to-End Model |
| Integration of LLM with A* Path Optimization | ✓ | ✗ | ✗ | ✗ | ✗ |
| User-Defined Cost-Quality Weightage & Tradeoff | ✓ | ✗ | ✗ | ✗ | ✗ |
| Multimodality Support | ✓ | ✗ | ✗ | ✗ | ✗ |
| Continual Learning/Tool Updates | ✗ | ✓ | ✗ | ✗ | ✗ |
| Feedback-Based Retrying and Model Selection | ✓ | ✓ | ✓ | ✗ | ✗ |
| Single Pass Edit | ✗ | ✗ | ✗ | ✗ | ✓ |

Figure 6 compares CoSTA* with baselines across task complexities. While it shows marginal improvement in simple tasks, its advantage becomes pronounced in complex tasks (3+ subtasks), attributed to its path optimization and feedback integration. The radar plot confirms CoSTA*'s scalability and multimodal capabilities, handling both image-only and text+image tasks effectively.

**Pareto Optimality Analysis** The Pareto front (Figure 3) shows CoSTA*'s ability to balance cost and quality by adjusting $\alpha$. $\alpha = 2$ prioritizes cost, while $\alpha = 0$ maximizes quality. Baselines lack this flexibility and fall short of the Pareto front due to lower quality at comparable costs, demonstrating CoSTA*'s superior cost-quality optimization. These results are the average improvements over the entire dataset. The cost comparison of CoSTA* with the baselines is also available in Table 9.

**Qualitative Results** Figure 1 provides qualitative comparisons, illustrating CoSTA*'s ability to seamlessly handle multimodal tasks. Table 5 highlights its distinct advantages, including real-time feedback, dynamic heuristic adjustments, and LLM integration with A* search—features lacking in baselines. We also present a qualitative comparison between CoSTA* and the very recent **Gemini 2.0 Flash Preview Image Generation** on a few tasks from our benchmark, in which our methods exhibit significant advantages over Gemini. This comparison can be seen in Figure 11.

## 5.4 ABLATION STUDY

To understand the contribution of various components in CoSTA*, we conducted several ablation studies, summarized in Table 6. These studies evaluate the impact of real-time feedback integration, multimodality support, the Model Description Table (MDT), and the Tool Dependency Graph (TDG). Furthermore, **to ensure a fair and comprehensive comparison with baseline methods, additional ablation studies restricting the tools and subtasks available to CoSTA* to only those supported by the baselines are detailed in Appendix D**. These appendix studies demonstrate that the performance improvements of CoSTA* are not solely due to its broader toolset, but stem from its superior planning capabilities. Appendix D also analyzes contributions of individual high-level components like LLM-based planning versus A* search only, and impact of the cost-quality tradeoff mechanism.

Table 6: Impact of core components on CoSTA*'s performance.

| Configuration / Component Removed | Average Accuracy |
|---|---|
| **CoSTA* (Full Method)** | **0.94** |
| no Real-time Feedback $g(x)$ ($h(x)$ only) | 0.80 |
| no Multimodality Support (Image-only tools for text tasks) | 0.48 |
| no Model Description Table | 0.85 |
| no Tool Dependency Graph (TDG) | 0.82 |

**Feedback Integration with $g(x)$** To isolate the impact of real-time feedback ($g(x)$), we compared our method against a variant relying solely on static heuristics ($h(x)$-only). Static heuristics may not always capture optimal tool choices in diverse scenarios. As shown in Table 6, the full CoSTA* method, by integrating $g(x)$ to adapt to actual tool performance, significantly boosted accuracy compared to the $h(x)$-only approach. An illustrative case, where a path guided by $h(x)$ alone is suboptimal but is effectively corrected by the full CoSTA* with $g(x)$ integration, is depicted in Figure 7 (bottom). This confirms that real-time feedback substantially enhances path selection and robustness within our framework.

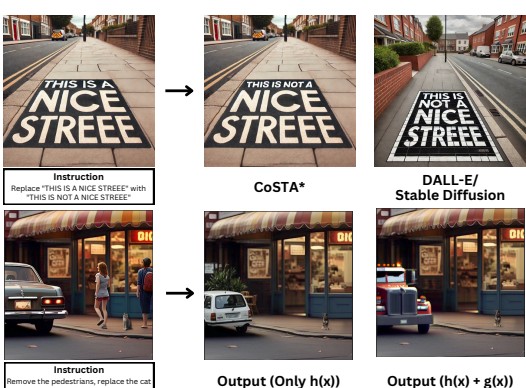

Figure 7: Qualitative comparison of image editing tools vs. CoSTA* (top), highlighting multimodal advantages; Comparison of $h(x)$ vs. $h(x) + g(x)$ (bottom), demonstrating improved editing precision from real-time feedback

**Impact of Multimodality Support** Comparing CoSTA*'s full multimodal capabilities on text-related tasks against a version restricted to only image-modality tools (e.g., DALL-E) revealed a substantial accuracy drop (Table 6). CoSTA*'s integration of specialized text-focused tools ensures better visual and textual fidelity, leading to significantly improved results, as qualitatively shown in Figure 7 (right).

**Impact of Model Description Table (MDT)** The Model Description Table (MDT, detailed in Table 18) provides structured tool information (supported subtasks, inputs, outputs). Ablating the MDT—providing the LLM only with model names and requiring it to infer capabilities—noticeably decreased accuracy (Table 6). This underscores the MDT's role in guiding the LLM for accurate tool-subtask mapping and reducing planning errors.

**Impact of Tool Dependency Graph (TDG)** The Tool Dependency Graph (TDG, Figure 4), defines valid tool sequences. Removing the TDG and requiring the LLM to infer these input/output dependencies significantly lowered accuracy (Table 6). This highlights the TDG's importance for plan feasibility and efficiency by preventing exploration of invalid tool sequences, thus improving CoSTA*'s reliability.

## 6 CONCLUSIONS

In this paper, we present a novel image editing agent that leverages the capabilities of a large multimodal model as a planner combined with the flexibility of the A* algorithm to search for an optimal editing path, balancing the cost-quality tradeoff. Experimental results demonstrate that CoSTA*effectively handles complex, real-world editing queries with reliability while surpassing existing baselines in terms of image quality. We believe that this neurosymbolic approach is a promising direction toward more capable and reliable agents in the future.

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

# A  COMPARISON ON MAGICBRUSH AND EMU-EDIT BENCHMARKS

**Comparison on MagicBrush**    To further validate the effectiveness of CoSTA*, we conducted experiments on the MagicBrush (Zhang et al., 2024a) benchmark. This benchmark provides a standardized set of images and editing instructions for both single-turn and multi-turn image editing tasks.

As shown in Table 7, CoSTA* consistently outperforms all baseline methods across all reported metrics (L1↓, L2↓, CLIP-I↑, CLIP-T↑) for both single-turn and multi-turn settings. These superior results can be attributed to CoSTA*'s robust planning mechanism, which can leverage multiple tools for the same subtask, and its automatic quality check at each step. This allows CoSTA* to select the best performing tool for each subtask and dynamically recover if a chosen tool fails to produce a satisfactory result. In contrast, other methods are often limited to single models or, like GenArtist, may use a single predefined tool for each subtask. While GenArtist can revise a subtask if the LLM initially chooses an incorrect one, it lacks the flexibility to select from different tools for the same subtask or to switch tools if one underperforms. CoSTA*'s ability to evaluate and choose among multiple tool options for each step in the editing process leads to higher quality and more robust editing outcomes.

Table 7: Quantitative Comparison on MagicBrush with existing image editing methods. Multi-turn setting evaluates images that are iteratively edited on the previous source images in edit sessions. CoSTA* demonstrates superior performance across all metrics. However, the improvement margin over baselines is less pronounced here compared to our own benchmark, as MagicBrush contains fewer highly complex tasks and a smaller proportion of multi-turn tasks (max 3 turns vs. up to 8 in ours).

| Settings | Methods | L1↓ | L2↓ | CLIP-I↑ | CLIP-T↑ |
|---|---|---|---|---|---|
| Single-turn | Null Text Inversion (Mokady et al., 2022) | 0.0749 | 0.0197 | 0.8827 | 0.2737 |
| | HIVE (Zhang et al., 2024b) | 0.1092 | 0.0341 | 0.8519 | 0.2752 |
| | InstructPix2Pix (Brooks et al., 2023) | 0.1122 | 0.0371 | 0.8524 | 0.2764 |
| | MagicBrush (Zhang et al., 2024a) | 0.0625 | 0.0203 | 0.9332 | 0.2781 |
| | SmartEdit (Huang et al., 2023) | 0.0810 | - | 0.9140 | 0.3050 |
| | GenArtist (Wang et al., 2024b) | 0.0536 | 0.0147 | 0.9403 | 0.3129 |
| | **CoSTA* (Ours)** | **0.0512** | **0.0139** | **0.9465** | **0.3206** |
| Multi-turn | Null Text Inversion (Mokady et al., 2022) | 0.1057 | 0.0335 | 0.8468 | 0.2710 |
| | HIVE (Zhang et al., 2024b) | 0.1521 | 0.0557 | 0.8004 | 0.2673 |
| | InstructPix2Pix (Brooks et al., 2023) | 0.1584 | 0.0598 | 0.7924 | 0.2726 |
| | MagicBrush (Zhang et al., 2024a) | 0.0964 | 0.0353 | 0.8924 | 0.2754 |
| | GenArtist (Wang et al., 2024b) | 0.0858 | 0.0298 | 0.9071 | 0.3067 |
| | **CoSTA* (Ours)** | **0.0825** | **0.0281** | **0.9143** | **0.3102** |

**Comparison on Emu-Edit Benchmark**    We evaluated CoSTA on the Emu-Edit benchmark to test its generalization capabilities. As shown in Table 8, CoSTA achieves the highest accuracy, demonstrating its robust planning and execution framework.

Table 8: Performance comparison on the Emu-Edit benchmark.

| Method | Accuracy (Emu-Edit) |
|---|---|
| CoSTA* | **0.95** |
| GenArtist | 0.81 |
| VisProg | 0.70 |
| CLOVA | 0.72 |
| Instruct Pix2Pix | 0.64 |
| MagicBrush | 0.68 |

# B  DETAILED NOVELTIES OF COSTA$^*$

This section provides a more detailed breakdown of the key novelties and contributions of the CoSTA$^*$ framework.

- **Hierarchical Planning with LLM and A* Synergy:**
  - A primary novelty is the integration of LLM-based high-level planning with a low-level A* graph search. This synergy leverages the LLM's strength in commonsense reasoning for subtask decomposition and search space pruning, while the A* search excels at finding optimal toolpaths within the pruned graph, handling complex workflows and numerical evaluations where LLMs might falter.
  - This hierarchical approach mitigates the weaknesses of using either method in isolation: LLMs alone can struggle with detailed, multi-tool planning and precise cost/quality estimation, while A* search alone on a full tool graph can be computationally intractable for complex tasks.

- **Advanced Tool Selection within Subtasks:**
  - Unlike methods that use a single, predefined tool per subtask, CoSTA$^*$'s planning method allows for dynamic selection of the most suitable tool from multiple available options for *each* subtask instance.
  - This selection is not solely based on pre-defined heuristics but also considers real-time execution data (actual cost and quality), allowing CoSTA$^*$ to choose a better-performing tool even if another tool successfully completes the subtask but with a suboptimal cost-quality outcome for the current specific case.

- **Dynamic Cost-Quality Trade-off and Optimization:**
  - CoSTA$^*$ introduces a sophisticated mechanism for balancing execution cost and output quality, a crucial aspect often overlooked in prior image editing agents.
  - We employ a novel formulation for both heuristic ($h(x)$) and actual execution ($g(x)$) costs, which dynamically incorporates both time and quality metrics. This allows for nuanced decision-making and can achieve significant cost reductions (up to 20% in experiments) when cost is prioritized.
  - The framework includes a tunable coefficient ($\alpha$) that allows users to explicitly define their preference for the cost-quality trade-off, leading to versatile solutions on the Pareto front.

- **Real-time Feedback and Adaptive Planning:**
  - Each subtask's output is evaluated by a Vision-Language Model (VLM).
  - If a tool fails or produces low-quality output, CoSTA$^*$ not only attempts retries but also updates its internal cost and quality estimates for that tool-subtask pair. This adaptive learning allows the A* search to quickly recover from failures and explore alternative, more promising toolpaths.

- **Principled Use of Prior Knowledge (Benchmark Table):**
  - CoSTA$^*$ systematically collects and utilizes benchmark performance data (execution time and quality scores) for various tools across different subtasks.
  - This curated Benchmark Table (BT) serves as the foundation for initializing the heuristic scores ($h(x)$) used in the A* search, providing empirically grounded guidance for tool selection from the outset. This is a novel approach to leveraging prior tool knowledge in an agentic framework.

- **Comprehensive Multimodality Support:**
  - CoSTA$^*$ is designed to handle complex tasks that require seamless integration of both image and text editing tools.
  - The framework can automatically switch between modalities across different subtasks within a single editing workflow, optimizing for the best cost-quality trade-off by selecting the most appropriate tool, regardless of its modality.

- **Novel Benchmark for Complex Multi-Turn Editing:**
  - We contribute a new, challenging benchmark specifically designed for evaluating multi-turn image editing agents. This benchmark includes tasks of varying complexities, with a higher proportion of multi-turn scenarios (up to 8 turns) compared to some existing benchmarks, facilitating more rigorous evaluation of sophisticated planning and execution capabilities.

## C  QUANTITATIVE COST COMPARISON

**Cost-Quality Trade-off Comparison**   The results in Table 9 provide a detailed breakdown of the cost (in seconds) and quality scores for CoSTA at different $\alpha$ values compared to baseline methods, reinforcing the Pareto optimality analysis shown in Figure 3.

Table 9: Detailed cost-quality comparison of CoSTA with baseline models.

| Method | Quality Score | Execution Cost (seconds) |
|---|---|---|
| **Ours** ($\alpha = 0$) | **0.956** | 67.7 |
| **Ours** ($\alpha = 0.5$) | 0.949 | 65.1 |
| **Ours** ($\alpha = 1$) | 0.927 | 58.2 |
| **Ours** ($\alpha = 1.5$) | 0.902 | 54.2 |
| **Ours** ($\alpha = 2$) | 0.889 | **53.1** |
| CLOVA | 0.570 | 61.2 |
| GenArtist | 0.862 | 65.7 |
| Instruct Pix2Pix | 0.520 | 67.2 |
| MagicBrush | 0.880 | 65.1 |

## D  ADDITIONAL ABLATION STUDIES

To provide a strictly fair and comprehensive comparison, and to further analyze the contribution of different components of CoSTA$^*$, we conducted several additional ablation studies.

### D.1  FAIR COMPARISON WITH RESTRICTED TOOLSET AND SUBTASKS

While the ability of CoSTA$^*$ to support a wider range of tools and subtasks is a notable advantage, we performed ablation studies where the toolset and supported subtasks for CoSTA$^*$ were restricted to only those available to the baseline methods (VisProg  (Gupta & Kembhavi, 2023), CLOVA  (Gao et al., 2024), and GenArtist  (Wang et al., 2024b)). This ensures that any observed performance difference is primarily due to the planning and execution strategy rather than the breadth of available tools.

As shown in Table 10 and Table 11, when the number of subtasks per task is low (1-2 subtasks), CoSTA$^*$ achieves a relatively marginal improvement (approximately 1.7% compared to the average of VisProg and CLOVA, and 0.4% compared to GenArtist). However, as the task complexity increases (e.g., 7-8 subtasks), the performance advantage of CoSTA$^*$ becomes significantly more pronounced, reaching approximately 33.5% against VisProg/CLOVA and 23.9% against GenArtist. This substantial improvement in complex scenarios underscores the superior planning and decision-making capabilities of CoSTA$^*$, which can efficiently navigate intricate multi-step editing tasks where baseline methods often struggle or fail. These results verify that our superior performance is primarily due to the method's design.

Table 10: Fair comparison of CoSTA$^*$ with VisProg and CLOVA using a restricted toolset and subtask list. Performance difference (Diff. w/ CoSTA$^*$) is calculated as the average of baselines vs. CoSTA$^*$.

| Subtasks | CoSTA$^*$ | VisProg (Gupta & Kembhavi, 2023) | CLOVA (Gao et al., 2024) | Diff. w/ CoSTA$^*$ |
|---|---|---|---|---|
| 1-2 Subtasks | **0.510** | 0.498 | 0.504 | -1.7% |
| 3-4 Subtasks | **0.551** | 0.489 | 0.496 | -10.6% |
| 5-6 Subtasks | **0.573** | 0.446 | 0.451 | -21.7% |
| 7-8 Subtasks | **0.460** | 0.301 | 0.310 | -33.5% |
| Overall Accuracy | **0.525** | 0.436 | 0.446 | -16.0% |

We also re-evaluated CoSTA$^*$ by reducing its supported subtasks to those only supported by all baselines collectively for a broader comparison. As shown in Table 12, our performance advantage is approximately 9% on simpler tasks but increases to nearly 47% on more complex ones, further confirming that the design of CoSTA$^*$ is key to its superior performance.

Table 11: Fair comparison of CoSTA* with GenArtist using a restricted toolset and subtask list. Performance difference (Diff. w/ CoSTA*) is calculated as GenArtist vs. CoSTA*.

| Subtasks | CoSTA* | GenArtist (Wang et al., 2024b) | Diff. w/ CoSTA* |
|---|---|---|---|
| 1-2 Subtasks | **0.508** | 0.506 | -0.39% |
| 3-4 Subtasks | **0.578** | 0.548 | -5.21% |
| 5-6 Subtasks | **0.606** | 0.503 | -17.11% |
| 7-8 Subtasks | **0.515** | 0.392 | -23.89% |
| Overall Accuracy | **0.553** | 0.495 | -10.55% |

Table 12: Fair comparison of CoSTA* with all baselines using a restricted subtask list common to all while using our own toolset for these corresponding allowed subtasks. Difference with CoSTA* is calculated as the average of all other baselines vs. CoSTA*.

| Subtasks | CoSTA* | VisProg (Gupta & Kembhavi, 2023) | CLOVA (Gao et al., 2024) | InstructPix2Pix (Brooks et al., 2023) | MagicBrush (Zhang et al., 2024a) | GenArtist (Wang et al., 2024b) | Difference with CoSTA* |
|---|---|---|---|---|---|---|---|
| 1-2 Subtasks | **0.550** | 0.498 | 0.504 | 0.497 | 0.501 | 0.506 | -8.87% |
| 3-4 Subtasks | **0.632** | 0.489 | 0.496 | 0.477 | 0.503 | 0.548 | -20.75% |
| 5-6 Subtasks | **0.654** | 0.446 | 0.451 | 0.390 | 0.402 | 0.503 | -32.96% |
| 7-8 Subtasks | **0.599** | 0.301 | 0.310 | 0.274 | 0.315 | 0.392 | -46.85% |
| Overall Accuracy | **0.610** | 0.436 | 0.446 | 0.413 | 0.434 | 0.495 | -27.08% |

## D.2 IMPACT OF CORE COMPONENTS

**CoSTA* without A\* Search (LLM-Only):** To evaluate the contribution of the A* search component, we compared CoSTA* with an LLM-only approach where the LLM is responsible for both high-level subtask planning and low-level tool selection without the refinement of A* search. The results in Table 13 clearly indicate that CoSTA*, by integrating A* search, significantly outperforms the LLM-only approach. This demonstrates that while LLMs are useful for high-level planning and pruning the search space, they struggle with the intricacies of selecting optimal tools from a large set, managing dependencies, incorporating heuristic costs, and handling failures robustly. These results were obtained by providing all benchmark data and detailed tool information (dependencies, inputs/outputs) directly to the LLM. If the LLM were to rely solely on its pre-existing knowledge base without this explicit information, the performance would likely degrade further.

Table 13: Comparison of CoSTA* with an LLM-only planning approach.

| Approach | Accuracy |
|---|---|
| LLM-only Approach | 0.73 |
| **CoSTA*** | **0.94** |

**CoSTA* without LLM for Planning (A\* Search Only):** Conversely, we examined the scenario where LLM-based high-level planning is removed, and A* search operates on a much larger, unpruned tool graph. In this A*-search-only method, the size of the search tree can grow exponentially with the number of subtasks, potentially involving over 100,000 nodes for complex tasks. This makes traversal and finding an optimal path computationally prohibitive and inefficient. In contrast, CoSTA* with LLM-based pruning effectively manages this complexity, typically maintaining only about 15-20 nodes in the active search queue. This highlights the critical role of the LLM in making the A* search feasible and efficient for complex multi-turn editing.

**CoSTA* without Cost/Quality Tradeoff:** We also analyzed the impact of the cost-quality tradeoff mechanism by evaluating CoSTA* when optimizing solely for cost ($\alpha = 2$), solely for quality ($\alpha = 0$), and with a balanced approach ($\alpha = 1$). The results are presented in Table 14.

These results demonstrate the importance of the cost-quality tradeoff in CoSTA*. Optimizing solely for one criterion (e.g., only cost) can lead to a noticeable compromise in the other (e.g., quality), and vice-versa. The ability to balance these factors via the $\alpha$ parameter allows CoSTA* to adapt to different user preferences and resource constraints effectively.

## E  STEP-BY-STEP EXECUTION OF TASKS IN FIGURE 1

To complement the qualitative comparisons presented in Figure 1, Figure 8 provides a visualization of the step-by-step execution of selected subtasks within the composite task by CoSTA*. This figure

Table 14: Impact of the cost-quality tradeoff parameter $\alpha$ in CoSTA*.

| Focus | Only Cost ($\alpha = 2$) | Only Quality ($\alpha = 0$) | Both ($\alpha = 1$) |
|---|---|---|---|
| Quality Score | 0.881 | 0.956 | 0.937 |
| Cost (in seconds) | 53.1 | 67.7 | 58.2 |

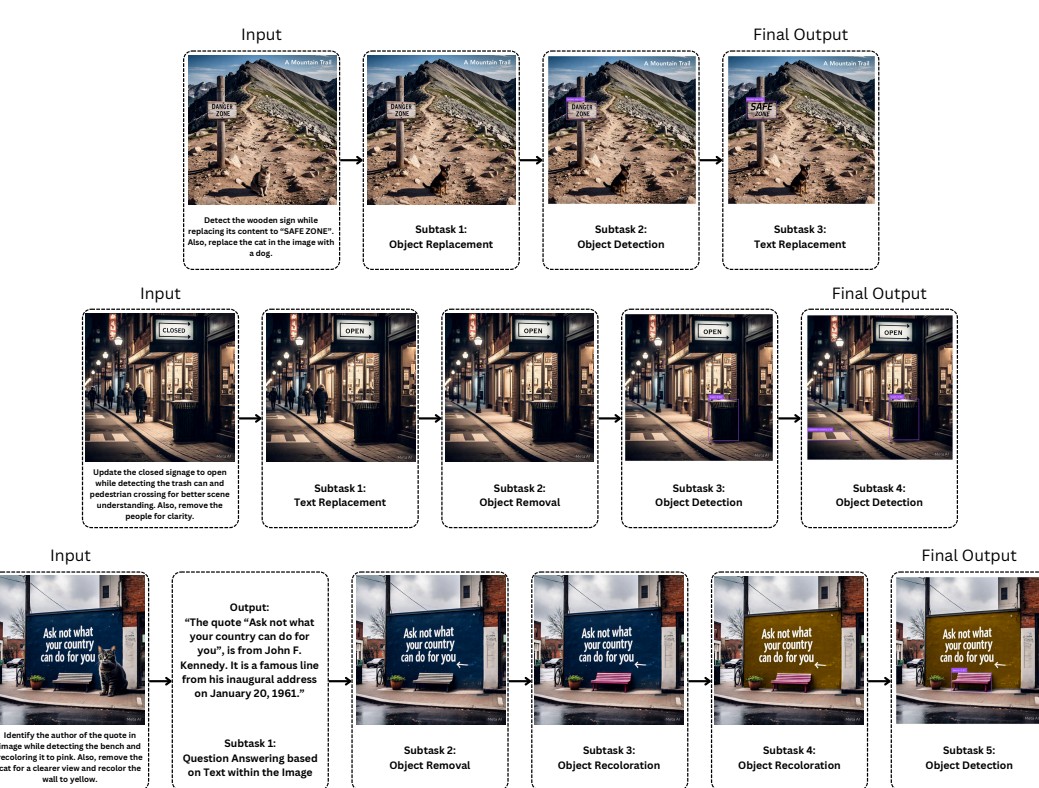

Figure 8: Step-by-step execution of editing tasks using CoSTA*. Each row illustrates an input image, the corresponding subtask breakdown, and intermediate outputs at different stages of the editing process. This visualization highlights how CoSTA* systematically refines outputs by leveraging specialized models for each subtask, ensuring greater accuracy and consistency in multimodal tasks.

highlights the intermediate outputs produced by each subtask, illustrating how complex image editing operations are decomposed and executed sequentially.

By showcasing the incremental progression of subtasks, this visualization provides a clearer view of how different intermediate outputs contribute to the final edited image. Rather than illustrating the full decision-making process of CoSTA*, the figure focuses on the stepwise transformations applied to the image, offering a practical reference for understanding the effects of each subtask.

This breakdown highlights key transitions in tasks, demonstrating the intermediate results generated at various stages. It provides insight into how each operation modifies the image, helping to better interpret the qualitative comparisons presented in the main text.

## F  HUMAN EVALUATION FOR ACCURACY CALCULATION

To ensure reliable performance assessment, we conduct human evaluations for accuracy calculation across all subtasks and tasks. Unlike automatic metrics such as CLIP similarity, human evaluation accounts for nuanced errors, semantic inconsistencies, and multi-step dependencies that are often missed by automated tools. We recruited a panel of 5 human evaluators. The inter-rater consistency was high, with a variance of 0.07, indicating objective adherence to the rubric. We emphasize that CLIP is used solely as a reward signal in the loop, while final claims rely entirely on this human

evaluation. This section outlines the evaluation methodology, scoring criteria, and aggregation process.

Table 15: Predefined Rules for Assigning Partial Correctness Scores in Human Evaluation

| Task Type | Evaluation Criteria | Assigned Score |
|---|---|---|
| Image-Only Tasks | Minor artifacts, barely noticeable distortions | 0.9 |
| | Some visible artifacts, but main content is unaffected | 0.8 |
| | Noticeable distortions, but retains basic correctness | 0.7 |
| | Significant artifacts or blending issues | 0.5 |
| | Major distortions or loss of key content | 0.3 |
| | Output is almost unusable, but some attempt is visible | 0.1 |
| Text+Image Tasks | Text is correctly placed but slightly misaligned | 0.9 |
| | Font or color inconsistencies, but legible | 0.8 |
| | Noticeable alignment or formatting issues | 0.7 |
| | Some missing or incorrect words but mostly readable | 0.5 |
| | Major formatting errors or loss of intended meaning | 0.3 |
| | Text placement is incorrect, missing, or unreadable | 0.1 |

## F.1 SUBTASK-LEVEL ACCURACY

Each subtask $s_i$ in a task $T$ is manually assessed by evaluators and assigned a correctness score $A(s_i)$ based on the following criteria:

$$A(s_i) = \begin{cases} 1, & \text{if the subtask is fully correct} \\ x, & \text{if the subtask is partially correct, where } x \in \{0.1, 0.3, 0.5, 0.7, 0.8, 0.9\} \\ 0, & \text{if the subtask has failed} \end{cases} \quad (5)$$

Partial correctness ($x$) is determined based on predefined task-specific criteria. Table 15 defines the rules used to assign these scores across different subtasks.

## F.2 TASK-LEVEL ACCURACY

Task accuracy is computed as the mean correctness of its subtasks:

$$A(T) = \frac{1}{|S_T|} \sum_{i=1}^{|S_T|} A(s_i) \quad (6)$$

where $S_T$ is the set of subtasks in task $T$, ensuring that task accuracy reflects overall subtask correctness.

## F.3 OVERALL ACCURACY ACROSS TASKS

To evaluate system-wide performance, the overall accuracy is computed as the average of task-level accuracies:

$$A_{\text{overall}} = \frac{1}{|T|} \sum_{j=1}^{|T|} A(T_j) \quad (7)$$

where $|T|$ is the total number of evaluated tasks.

## G AUTOMATIC CONSTRUCTION OF THE TOOL DEPENDENCY GRAPH

The Tool Dependency Graph (TDG) can be automatically generated by analyzing the input-output relationships of each tool. Each tool $v_i$ is associated with a set of required inputs $\mathcal{I}(v_i)$ and a set of produced outputs $\mathcal{O}(v_i)$. We construct directed edges $(v_i, v_j)$ whenever $\mathcal{O}(v_i) \cap \mathcal{I}(v_j) \neq \emptyset$, meaning the output of tool $v_i$ is required as input for tool $v_j$.

These input-output relationships are explicitly listed in the **Model Description Table (MDT)**, where two dedicated columns specify the expected inputs and produced outputs for each tool. Using this structured metadata, the TDG can be dynamically constructed without manual intervention, ensuring that dependencies are correctly captured and automatically updated as the toolset evolves.

# H    DATASET GENERATION AND EVALUATION SETUP

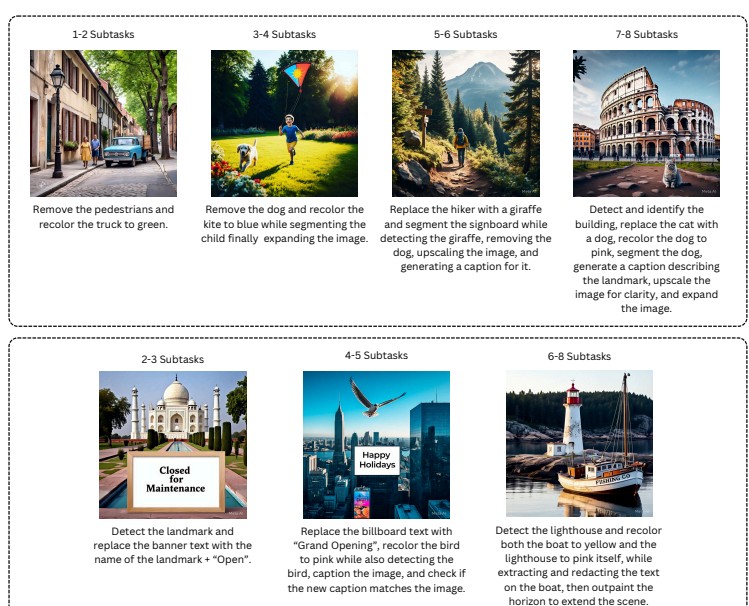

Figure 9: An overview of the dataset used for evaluation, showcasing representative input images and prompts across different task categories. The top section presents examples from image-only tasks, while the bottom section includes text+image tasks. These examples illustrate the diversity of tasks in our dataset, highlighting the range of modifications required for both visual and multimodal editing scenarios.

## H.1    DATASET CONSTRUCTION FOR BENCHMARKING

To rigorously evaluate the effectiveness of our method, we constructed a diverse, large-scale dataset designed to test various image editing tasks under complex, multi-step, and multimodal constraints. The dataset generation process was carefully structured to ensure both realism and consistency in task complexity.

### H.1.1    AUTOMATIC PROMPT GENERATION & HUMAN CURATION

To simulate real-world image editing tasks, we first generated a diverse set of structured prompts using a **Large Language Model (LLM)**. These prompts were designed to cover a wide variety of editing operations, including:

- Object **replacement, addition, removal, and recoloration**,
- **Text-based modifications** such as replacement, addition, and redaction,
- **Scene-level changes**, including background modification and outpainting.

While LLM-generated prompts provided an automated way to scale dataset creation, they lacked real-world editing constraints. Thus, each prompt was manually curated by human annotators to ensure:

1. **Logical Feasibility:** Ensuring that edits could be performed realistically on an image.

2. **Complexity Diversity:** Creating **simple (1-2 subtasks)** and **complex (5+ subtasks)** tasks for a comprehensive evaluation.

3. **Ensuring Clarity:** Refining ambiguous phrasing or vague instructions.

### H.1.2 IMAGE GENERATION WITH META AI

Once the curated prompts were finalized, **image generation** was performed using **Meta AI's generative model**. Unlike generic image generation, our **human annotators provided precise instructions** to ensure that:

- **Every key element mentioned in the prompt** was included in the generated image.
- The **scene, object attributes, and text elements** were visually clear for the intended edits.
- The images had sufficient complexity and diversity to challenge different image-editing models.

For example, if a prompt requested *"Replace the red bicycle with a blue motorcycle and remove the tree in the background,"* the generated image explicitly contained a **red bicycle and a clearly distinguishable tree**, ensuring that subsequent edits could be precisely evaluated.

### H.2 DATASET COMPOSITION & SUBTASK DISTRIBUTION

Our dataset comprises **121 total image-task pairs**, with tasks spanning both **image-only** and **text+image** categories. Each image-editing prompt is decomposed into **subtasks**, which are then mapped to the supported models for evaluation.

Figure 10: Distribution of the number of instances for each subtask in the dataset.

Figure 10 illustrates the distribution of subtasks across the dataset. This provides insights into:

- The **relative frequency** of each subtask.
- The **balance between different categories** (e.g., object-based, text-based, scene-based).

The dataset ensures adequate representation of each subtask, avoiding skew toward a specific category. The most common subtasks in the dataset include **Object Replacement, Object Recoloration, and Object Removal**, while rarer but complex operations like **Keyword Highlighting** remain crucial for evaluation.

# I    COMPARISON WITH RECENT CLOSED-SOURCE IMAGE EDITING MODELS

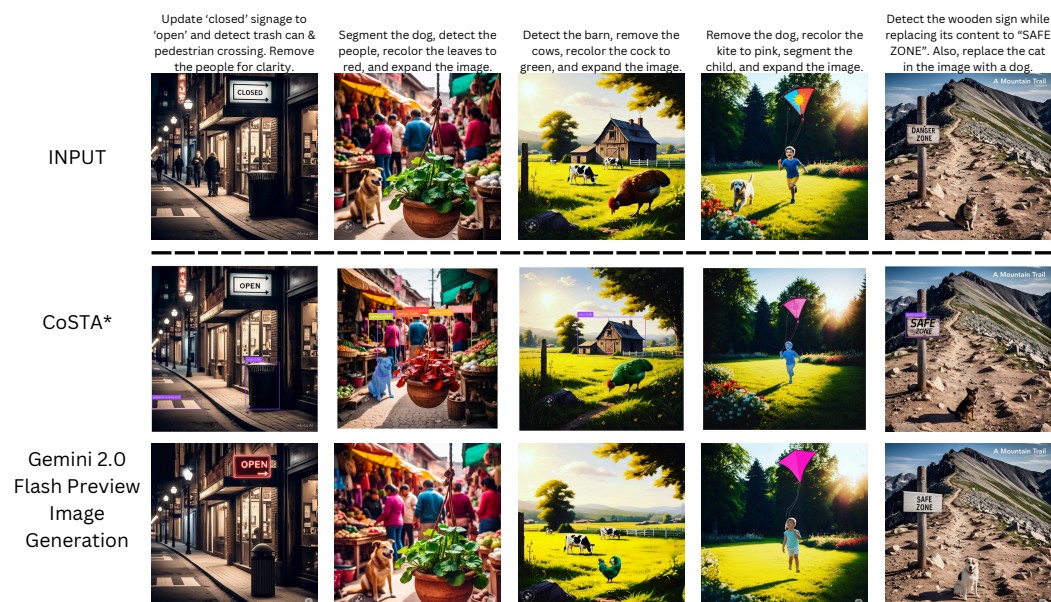

Figure 11: Comparison of CoSTA* with the Gemini 2.0 Flash Preview Image Generation on a few tasks from our benchmark. These examples highlight CoSTA*'s effectiveness in precisely handling diverse operations like text manipulation, object replacement, etc., often with greater adherence to the detailed instructions compared to Gemini.

**Comparison with Gemini and GPT-4o**    We conducted further quantitative comparisons on our evaluation benchmark with both Gemini 2.0 Flash and GPT-4o (with image editing). The results, summarized in Table 16, show that CoSTA's structured, tool-based approach yields a significant advantage in quality over these large generative models for complex editing tasks.

Table 16: Quantitative comparison with Gemini 2.0 and GPT-4o on our benchmark.

| Method | Average Quality Score |
|---|---|
| **Ours** | **0.94** |
| Gemini 2.0 | 0.81 |
| GPT-4o (with image editing) | 0.78 |

Table 17: Average CLIP Similarity Scores for Outputs of Randomness-Prone Subtasks

| Subtask | Avg CLIP Score |
|---|---|
| Object Replacement | 0.98 |
| Object Recoloration | 0.99 |
| Object Addition | 0.97 |
| Object Removal | 0.97 |
| Image Captioning | 0.92 |
| Outpainting | 0.99 |
| Changing Scenery | 0.96 |
| Text Removal | 0.98 |
| QA on Text | 0.96 |

## J  CONSISTENCY IN COSTA* OUTPUTS

To assess robustness against randomness, we evaluated CoSTA* on subtasks prone to variability, such as object replacement and recoloration, where outputs may slightly differ across executions (e.g., different dog appearances when replacing a cat). A set of 20 images per subtask was selected, and each was processed multiple times. Outputs for each image were compared among each other using CLIP similarity scores, measuring consistency. As summarized in Table 17, CoSTA* maintains high similarity across runs, confirming its stability. Variability was negligible in most cases, except for image captioning (0.92 similarity), where multiple valid descriptions naturally exist. These results demonstrate that CoSTA* is highly consistent, with minimal impact from randomness.

Table 18: Model Description Table (MDT). Each model is listed with its supported subtasks, input dependencies, and outputs.

| Model | Tasks Supported | Inputs | Outputs |
|---|---|---|---|
| Grounding DINO (Liu et al., 2024) | Object Detection | Input Image | Bounding Boxes |
| YOLOv7 | Object Detection | Input Image | Bounding Boxes |
| SAM (Kirillov et al., 2023b) | Object Segmentation | Bounding Boxes | Segmentation Masks |
| DALL-E | Object Replacement | Segmentation Masks | Edited Image |
| DALL-E | Text Removal | Text Region Bounding Box | Image with Removed Text |
| Stable Diffusion Erase | Text Removal | Text Region Bounding Box | Image with Removed Text |
| Stable Diffusion Inpaint | Object Replacement, Object Recoloration, Object Removal | Segmentation Masks | Edited Image |
| Stable Diffusion Erase | Object Removal | Segmentation Masks | Edited Image |
| Stable Diffusion 3 | Changing Scenery | Input Image | Edited Image |
| Stable Diffusion Outpaint | Outpainting | Input Image | Expanded Image |
| Stable Diffusion Search & Recolor | Object Recoloration | Input Image | Recolored Image |
| Stable Diffusion Remove Background | Background Removal | Input Image | Edited Image |
| Text Removal (Painting) | Text Removal | Text Region Bounding Box | Image with Removed Text |
| DeblurGAN (Kupyn et al., 2018) | Image Deblurring | Input Image | Deblurred Image |
| LLM (GPT-4o) | Image Captioning | Input Image | Image Caption |
| LLM (GPT-4o) | Question Answering based on text, Sentiment Analysis | Extracted Text, Font Style Label | New Text, Text Region Bounding Box, Text Sentiment, Answers to Questions |
| Google Cloud Vision (Google Cloud, 2024) | Landmark Detection | Input Image | Landmark Label |
| CRAFT (Baek et al., 2019) | Text Detection | Input Image | Text Bounding Box |
| CLIP (Radford et al., 2021) | Caption Consistency Check | Extracted Text | Consistency Score |
| DeepFont (Wang et al., 2015) | Text Style Detection | Text Bounding Box | Font Style Label |
| EasyOCR (Kittinaradorn et al., 2022) | Text Extraction | Text Bounding Box | Extracted Text |
| MagicBrush (Zhang et al., 2024a) | Object Addition | Input Image | Edited Image with Object |
| pix2pix (Isola et al., 2018) | Changing Scenery | Input Image | Edited Image |
| Real-ESRGAN (Wang et al., 2021) | Image Upscaling | Input Image | High-Resolution Image |
| Text Writing using Pillow | Text Addition | New Text, Text Region Bounding Box | Image with Text Added |
| Text Writing using Pillow | Text Replacement, Keyword Highlighting | Image with Removed Text | Image with Text Added |
| Text Redaction (Code-based) | Text Redaction | Text Region Bounding Box | Image with Redacted Text |
| MiDaS (Ranftl et al., 2020) | Depth Estimation | Input Image | Image with Depth of Objects |

## K  MODEL DESCRIPTION TABLE (MDT)

The full Model Description Table (MDT) provides a comprehensive list of all 22 specialized models used in the CoSTA* pipeline for image and text-in-image editing. Each model is mapped to its supported subtasks, input dependencies, and outputs, ensuring optimal tool selection for diverse editing requirements. These structured input-output relationships enable the automatic construction of the **Tool Dependency Graph (TDG)** by identifying dependencies between models based on their required inputs and generated outputs. Unlike generic pipelines, CoSTA* utilizes targeted models to enhance accuracy and efficiency in text-related visual tasks. Table 18 presents the complete MDT, detailing the capabilities of each model across different task categories and their role in facilitating automated dependency resolution.

## L  BENCHMARK TABLE (BT)

The Benchmark Table (BT) defines execution time and accuracy scores for each tool-task pair $BT(v_i, s_j)$, where $v_i$ is a tool and $s_j$ is a subtask. It serves as a baseline for A* search, enabling efficient tool selection. Both execution time and accuracy scores are based on empirical evaluations and published benchmarks (wherever available). For tools without prior benchmarks, evaluations on 137 instances of the specific subtask were conducted on 121 images from the dataset, with results included in Table 19. Accuracy values are normalized with respect to max within each subtask on a [0,1] scale for comparability.

Table 19: Benchmark Table for Accuracy and Execution Time. Accuracy and execution time for each tool-task pair are obtained from cited sources where available. For tools without prior benchmarks, evaluation was conducted over **137 instances of the specific subtask on 121 images from the dataset**, ensuring a robust assessment across varied conditions. Manual evaluation refers to our own evaluations on 137 instances of this subtask. The accuracy values for all models within a subtask are normalized with respect to max. The rationale for normalizing these accuracy scores is explained in Appendix T

| Model Name | Subtask | Accuracy (Normalized within Subtask) | Time (s) | Source |
|---|---|---|---|---|
| DeblurGAN (Kupyn et al., 2018) | Image Deblurring | 1.00 | 0.8500 | (Kupyn et al., 2018) |
| MiDaS (Ranftl et al., 2020) | Depth Estimation | 1.00 | 0.7100 | Manual |
| YOLOv7 (Wang et al., 2022) | Object Detection | 0.82 | 0.0062 | (Wang et al., 2022) |
| Grounding DINO (Liu et al., 2024) | Object Detection | 1.00 | 0.1190 | Accuracy: (Liu et al., 2024), Time: Manual |
| CLIP (Radford et al., 2021) | Caption Consistency Check | 1.00 | 0.0007 | Manual |
| SAM (Ravi et al., 2024) | Object Segmentation | 1.00 | 0.0500 | Accuracy: Manual, Time: (Ravi et al., 2024) |
| CRAFT (Baek et al., 2019) | Text Detection | 1.00 | 1.2700 | Accuracy: (Baek et al., 2019), Time: Manual |
| Google Cloud Vision (Google Cloud, 2024) | Landmark Detection | 1.00 | 1.2000 | Manual |
| EasyOCR (Kittinaradorn et al., 2022) | Text Extraction | 1.00 | 0.1500 | Manual |
| Stable Diffusion Erase | Object Removal | 1.00 | 13.8000 | Manual |
| DALL-E | Object Replacement | 1.00 | 14.1000 | Manual |
| Stable Diffusion Inpaint | Object Removal | 0.93 | 12.1000 | Manual |
| Stable Diffusion Inpaint | Object Replacement | 0.97 | 12.1000 | Manual |
| Stable Diffusion Inpaint | Object Recoloration | 0.89 | 12.1000 | Manual |
| Stable Diffusion Search & Recolor | Object Recoloration | 1.00 | 14.7000 | Manual |
| Stable Diffusion Outpaint | Outpainting | 1.00 | 12.7000 | Manual |
| Stable Diffusion Remove Background | Background Removal | 1.00 | 12.5000 | Manual |
| Stable Diffusion 3 | Changing Scenery | 1.00 | 12.9000 | Manual |
| pix2pix (Isola et al., 2018) | Changing Scenery (Day2Night) | 1.00 | 0.7000 | Accuracy: (Isola et al., 2018), Time: Manual |
| Real-ESRGAN (Wang et al., 2021) | Image Upscaling | 1.00 | 1.7000 | Manual |
| LLM (GPT-4o) | Question Answering based on Text | 1.00 | 6.2000 | Manual |
| LLM (GPT-4o) | Sentiment Analysis | 1.00 | 6.1500 | Manual |
| LLM (GPT-4o) | Image Captioning | 1.00 | 6.3100 | Manual |
| DeepFont (Wang et al., 2015) | Text Style Detection | 1.00 | 1.8000 | Manual |
| Text Writing - Pillow | Text Replacement | 1.00 | 0.0380 | Manual |
| Text Writing - Pillow | Text Addition | 1.00 | 0.0380 | Manual |
| Text Writing - Pillow | Keyword Highlighting | 1.00 | 0.0380 | Manual |
| MagicBrush (Zhang et al., 2023a) | Object Addition | 1.00 | 12.8000 | Accuracy: (Zhang et al., 2023a), Time: Manual |
| Text Redaction | Text Redaction | 1.00 | 0.0410 | Manual |
| Text Removal by Painting | Text Removal (Fallback) | 0.20 | 0.0450 | Manual |
| DALL-E (Ramesh et al., 2021) | Text Removal | 1.00 | 14.2000 | Manual |
| Stable Diffusion Erase (Rombach et al., 2022a) | Text Removal | 0.97 | 13.8000 | Manual |

# M    FAILURE CASE ANALYSIS AND LIMITATIONS

A key aspect of CoSTA*'s robustness is its sophisticated planning mechanism, featuring A* search and dynamic quality checks, designed to select the best available tool and recover from individual tool failures by exploring alternatives. However, the final output quality is also contingent upon the capabilities of the individual tools within its arsenal. There might be rare outlier cases or highly specialized subtasks for which no currently integrated tool can produce a satisfactory result. For instance, in the example conceptualized in Figure 12, CoSTA* correctly plans the sequence of operations for a recoloring task. When a tool in the initial path like SD Inpaint fails, CoSTA*'s A* search, guided by updated cost-quality metrics, explores alternatives such as SD Search&Recolor. Yet, even this alternative tool, despite being the next best option, cannot achieve a satisfactory outcome for the specific challenging case, and the final edited image for this subtask does not meet the desired quality. This illustrates that while our method intelligently navigates tool selection and failure recovery, its ultimate success in every conceivable scenario is bounded by the collective efficacy of the available tools. Furthermore, the overall effectiveness of CoSTA* also has a dependency on the initial task decomposition provided by the LLM; while the currently employed LLM handles our benchmark tasks well, exceptionally complex or ambiguous instructions beyond current LLM reasoning capacities might lead to suboptimal initial plans.

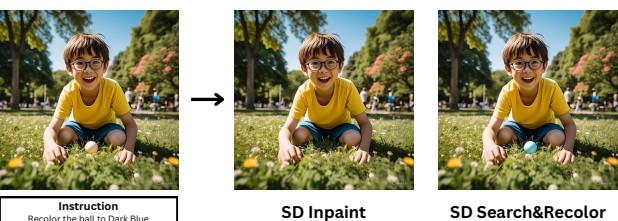

Figure 12: Example of a rare failure case where no available tool in the CoSTA* arsenal could satisfactorily complete a specific subtask, despite robust planning and retry mechanisms.

## N    CORRELATION ANALYSIS OF CLIP SCORES AND HUMAN ACCURACY

We analyzed the correlation between CLIP similarity scores and human accuracy across 40 tasks to assess CLIP's reliability in evaluating complex image-text edits. The scatter plot (Figure 13) illustrates the weak correlation, with Spearman's $\rho = 0.59$ and Kendall's $\tau = 0.47$, indicating that CLIP often fails to capture fine-grained inaccuracies. Despite assigning high similarity scores, CLIP struggles with detecting missing objects, distinguishing between multiple valid outputs, and recognizing context-dependent errors. Many instances where CLIP scored above 0.95 had human accuracy below 0.75, reinforcing the need for human evaluation in multimodal tasks. These findings highlight the limitations of CLIP as a standalone metric and emphasize the necessity of integrating human feedback for more reliable assessment.

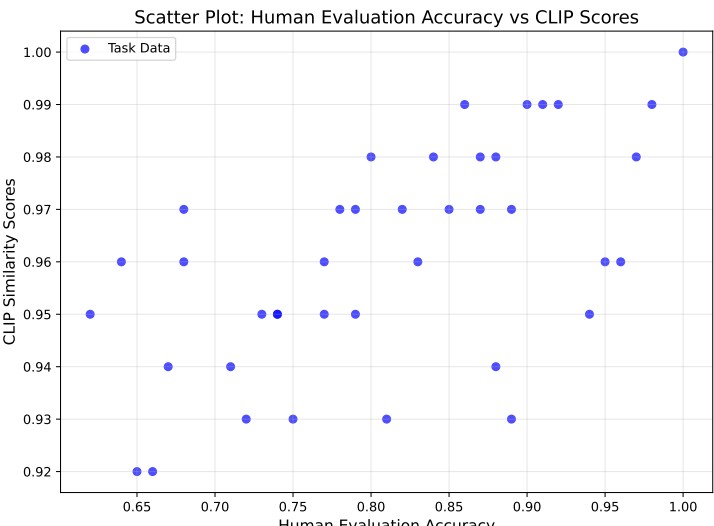

Figure 13: Scatter plot of CLIP scores vs. human accuracy across 40 tasks. The weak correlation (Spearman's $\rho = 0.59$, Kendall's $\tau = 0.47$) highlights CLIP's limitations in capturing nuanced inaccuracies, particularly in complex, multi-step tasks.

## O    A* EXECUTION STRATEGY

CoSTA$^*$ initializes heuristic values using benchmark data and dynamically updates execution costs based on real-time performance. The A$^*$ search iteratively selects the node with the lowest $f(x)$, explores its neighbors, and updates the corresponding values. If execution quality is below threshold, a retry mechanism adjusts parameters and re-evaluates $g(x)$ (Figure 14). The process continues until a leaf node is reached. By integrating precomputed heuristics with real-time cost updates, CoSTA$^*$ efficiently balances execution time and quality. This adaptive approach ensures robust decision-making, outperforming existing agentic and non-agentic baselines in complex multimodal editing tasks.

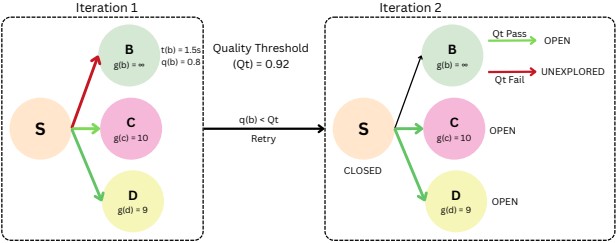

Figure 14: Visualization of the Retry Mechanism

## P    EMPIRICAL JUSTIFICATION OF HEURISTIC COST FORMULATION

The heuristic cost function $h(x)$ defined in Equation (3) was not selected arbitrarily but was developed through a rigorous empirical process designed to effectively balance execution cost with output quality. To validate this design, we evaluated four distinct mathematical formulations of the heuristic on a held-out validation set of tasks. The goal was to identify a formulation that remains stable across diverse tools while allowing controllable trade-offs via the $\alpha$ parameter.

We compared the following configurations:

- **Cost-Only** ($h(x) \approx C(y)$)**:** Prioritizes speed exclusively (similar to $\alpha = 2$).
- **Quality-Only** ($h(x) \approx 2 - Q(y)$)**:** Prioritizes quality exclusively (similar to $\alpha = 0$).
- **Linear Combination** ($h(x) = C(y)^\alpha - Q(y)^{2-\alpha}$)**:** An additive/subtractive approach attempting to balance both terms linearly.
- **Multiplicative (Ours, Eq. 3):** The proposed formulation combining terms multiplicatively: $[C(y)]^\alpha \times [2 - Q(y)]^{2-\alpha}$.

The results of this ablation are summarized in Table 20.

Table 20: Comparison of different heuristic formulations on a validation set. The Linear Combination proved unstable, often favoring low-quality tools due to low costs dominating the score. The proposed Multiplicative formulation achieves the best balance.

| Formulation | Validation Quality | Avg. Cost (s) | Observation |
|---|---|---|---|
| Cost-Only | 0.87 | **53.1** | Sacrifices quality for speed. |
| Quality-Only | **0.96** | 67.7 | Prohibitively high latency. |
| Linear Combination | 0.89 | Variable | Unstable; low-cost tools dominate. |
| **Multiplicative (Ours)** | 0.93 | 58.2 | **Optimal Balance & Stability** |

**Analysis:**

- The **Cost-Only** approach resulted in a significant drop in quality (0.87), as the agent consistently selected faster, lower-fidelity tools (e.g., lower-quality inpainting models) regardless of the visual outcome.
- The **Quality-Only** approach maximized visual fidelity (0.96) but led to a steep increase in execution cost ( 69s), making it unsuitable for time-sensitive agentic workflows.
- The **Linear Combination** proved mathematically unstable. We observed that tools with very low execution costs would disproportionately lower the $h(x)$ score, "tricking" the search into selecting them even if their quality score $Q(y)$ was poor. This lack of normalization between the two units (seconds vs. 0-1 score) made the trade-off uncontrollable.
- The **Multiplicative Formulation (Eq. 3)** effectively normalizes the disparate units of time and quality. It achieved a high quality score (0.93) with a moderate cost (58.2s). Crucially, it provides the mathematical stability required to enable the explicit, Pareto-optimal trade-offs shown in Figure 3 of the main text.

## Q    DETAILED MOTIVATIONS AND CONTEXT FOR AGENT-BASED PLANNING IN IMAGE EDITING

This section expands on the core motivations for employing an agentic, planning-based framework like CoSTA* for complex image editing. The need for such systems is increasingly recognized, with a growing body of work exploring agentic architectures to overcome the limitations of single-model, end-to-end generation (Wang et al., 2024b; Gupta & Kembhavi, 2023; Gao et al., 2024). We address why simple tool invocation is insufficient for this domain by discussing the challenges of compositionality, the necessity of iterative refinement, and by contextualizing our search-based approach with alternative planning paradigms.

## Q.1 The Challenge of Compositionality and Task Dependencies

A primary motivation for an agentic approach is that user instructions for image editing are rarely single, atomic actions. They are inherently **compositional**, often requiring a sequence of dependent operations to achieve the desired result. Recognizing this, recent state-of-the-art methods like GenArtist have been designed as agentic systems where a large multimodal model "coordinates various models to decompose intricate tasks into manageable sub-problems, enabling systematic planning".

For example, consider the prompt from our benchmark: "recolor the chalkboard to red while redacting the text on it and write 'A CLASSROOM' on the top.". A successful execution requires a specific order of operations:

1. **Text Detection & Redaction:** First, the existing text must be identified and removed from the chalkboard.
2. **Object Recoloration:** Only after the text is gone can the chalkboard's surface be cleanly recolored to red.
3. **Text Addition:** Finally, the new text can be written onto the newly colored surface.

This creates a natural **dependency graph** of subtasks that requires intelligent planning. This level of structured, sequential reasoning goes beyond the capabilities of earlier systems focused on simpler tool orchestration, such as VisProg (Gupta & Kembhavi, 2023), which lack the deep planning needed for highly compositional instructions. An intelligent agent is therefore essential for inferring this dependency graph from a natural language prompt and executing the subtasks in a valid order.

## Q.2 The Agentic Multiplier and Structural Efficiency

Optimizing inference time is critical because agentic workflows multiply the latency of base models (e.g., Detection → Segmentation → Inpainting). A single multi-turn task involves a chain of multiple tool calls, meaning any latency in the base tools is compounded, making the total execution time prohibitively high for interactive applications. Even if tools evolve (rendering specific costs ephemeral), the **efficiency gap** between methods is structural: CoSTA$^*$ finds intrinsically shorter and cheaper paths compared to greedy baselines. By reducing the number of necessary steps and retries, our approach also inherently reduces the total accumulated API network latency, achieving a $\sim 20\%$ reduction in total execution time compared to baselines.

## Q.3 The Imperative of Failure Recovery and Iterative Refinement

Generative AI tools are powerful but inherently stochastic and imperfect. Achieving high visual quality and fine-grained fidelity (e.g., in rendering textures or delicate structures like fingers) in a single generation step is a well-documented challenge. This has led to a paradigm shift towards multi-step refinement processes. Recent work, such as Interleaving Reasoning Generation (IRG), explicitly addresses this by proposing a framework that alternates between reasoning and generation to iteratively refine an image through reflection. Similarly, other agentic frameworks like GenArtist are built with "self-correction" capabilities to handle initial errors.

This recognized need for refinement and error correction motivates our agentic design. While IRG uses a learned model to "reflect" on an image, CoSTA* tackles this same fundamental challenge through its search-based agentic framework. Our approach provides a robust mechanism for iterative refinement driven by two key components:

1. **Real-time Quality Validation:** After each tool execution, a Vision-Language Model (VLM) performs a quality check to verify if the subtask was completed successfully.
2. **Dynamic Re-planning via A\* Search:** If the VLM check fails, the cost $g(x)$ associated with that failed toolpath is significantly increased. The A* search algorithm, always exploring the path with the lowest total estimated cost, naturally and immediately discards the failed path and pivots to explore the next-best alternative from its priority queue.

This allows the agent to dynamically recover from unexpected tool failures, providing a robust, search-based alternative to learned refinement models for achieving high-quality final outputs.

Q.4    Modularity, Extensibility, and Training-Efficiency

A significant, practical advantage of agent-based frameworks like CoSTA* is their inherent **training-efficiency and modularity**. Unlike monolithic end-to-end models that require extensive training on massive, domain-specific datasets to learn new capabilities, our agentic approach is comparatively inexpensive and far more adaptable.

**Training-Efficiency.**    Our framework leverages a collection of pre-existing, specialized models as tools. The core agent itself does not need to be trained from scratch on image editing tasks. It orchestrates these expert tools, harnessing their power without incurring the prohibitive computational cost of training a single, giant model to perform all functions. This makes the system lightweight and accessible, as it builds upon the collective progress of the open-source community rather than reinventing each capability.

**Ease of Extensibility.**    The modular "plug-and-play" architecture ensures that the system can be easily updated and extended. To support a new subtask or integrate a new, state-of-the-art tool, one does not need to retrain the entire system. Instead, the process is simple: the new tool is added to the Model Description Table and its performance metrics are added to the Benchmark Table. This declarative change is trivial compared to the significant engineering effort required for monolithic models, which would necessitate new data collection, architectural changes, and complete retraining on huge datasets. This flexibility ensures that the agent can remain current and powerful over time with minimal maintenance overhead.

Q.5    Comparison with Alternative Planning Paradigms: CATP-LLM

Recent works such as CATP-LLM (Wu et al., 2025) also explore cost-aware planning through a **learning-based approach**. However, CoSTA* differs from them by being a **search-based framework** which has several advantages for image editing domain:

- **No Fine-Tuning Required:** CoSTA* works out-of-the-box by leveraging a general LLM for high-level planning and A* search with pre-computed heuristics. In contrast, CATP-LLM requires extensive offline reinforcement learning on a large, generated dataset to fine-tune its policy model. Our approach is more lightweight, general, and easier to deploy.

- **Superior Dynamic Recovery:** Our A* framework provides more robust and efficient online adaptability. When a tool fails its quality check, the path's cost is updated, and the search immediately and naturally pivots to the next-best alternative in its priority queue. A pre-trained RL policy, like that in CATP-LLM, is less flexible in handling these real-time execution failures without more complex re-planning mechanisms.

- **Greater Extensibility:** Integrating new tools into CoSTA* is simple: one only needs to add them to the Benchmark Table and Tool Dependency Graph. A learned approach like CATP-LLM would likely require significant effort, including retraining its tool embeddings and fine-tuning the entire policy model to accommodate new tools.

In summary, CoSTA*'s search-based architecture offers a more practical, adaptable, and flexible solution for cost-sensitive planning in the dynamic domain of image editing.

# R    Dynamic Construction and Low Overhead of the Benchmark Table

A potential concern regarding our framework is that the Benchmark Table (BT), which provides heuristic scores for the A* search, represents a significant, manually-intensive prerequisite. In this section, we clarify that the BT is not a rigid overhead but rather a flexible component that can be constructed dynamically with minimal manual effort, ensuring the practicality and scalability of the CoSTA* framework.

### R.1 Dynamic Population via a "Cold Start" Inference Process

The CoSTA$^*$ framework does not strictly require a fully populated, hand-crafted Benchmark Table to function. Instead, it can be initialized using a **"cold start"** approach, making the BT an emergent property of the system's operation rather than a prerequisite.

The process is as follows:

1. **Initialization with Placeholders:** The system can begin with a naïve BT where all tool-task pairs are assigned generic, **placeholder values**. For example, all quality scores ($Q(v_i, s_j)$) can be initialized to a uniform value (e.g., 0.8), and all execution costs ($C(v_i, s_j)$) can be set to an average time derived from a few sample runs.

2. **Automatic Updates During Inference:** The core of our A* search is the real-time execution cost, $g(x)$, which is computed dynamically based on the actual performance of a tool on a given subtask. This function captures the true quality score and execution time observed during an inference pass.

3. **Convergence to a Stable BT:** By running the CoSTA* pipeline multiple times (e.g., 100-200 inference runs on a diverse set of tasks), the system naturally collects a rich set of these real-time performance data points. These observed values can then be aggregated (e.g., by averaging) to populate a new, empirically-grounded BT. The placeholder values are thus replaced with stable, realistic heuristics derived from the system's own experience.

This "inference-to-populate" mechanism demonstrates that the BT is not a static burden but can be learned and refined automatically over time, effectively eliminating the need for extensive, upfront manual experimentation.

### R.2 Semi-Automated Heuristic Collection for Faster Initialization

For users who wish to start with a more informed BT without a "cold start" phase, the collection of initial heuristic values can be largely automated, further reducing manual effort.

**Automating Cost Collection.** Many of the tools used in our pipeline are well-established open-source models. Execution costs (or reasonable estimates thereof) can often be found in their respective papers, repositories, or performance blogs. This information-gathering task can be delegated to a modern Large Language Model with internet search capabilities. By providing the LLM with a list of tools, it can be prompted to find and tabulate their typical execution times on standard hardware, providing a strong baseline for the cost heuristics with minimal human intervention.

**Minimizing Manual Quality Evaluation.** The most labor-intensive part of creating the BT is evaluating tool quality. However, extensive manual evaluation is only necessary in a specific scenario: when multiple tools compete to perform the *same* subtask. In this case, manual evaluation helps establish a relative performance ranking. For the many subtasks in our framework that are handled by a single, specialized tool, the quality score is normalized to 1.0 by default, requiring no manual evaluation. This targeted approach significantly reduces the scope of manual work to only a small subset of the toolset.

In summary, the Benchmark Table should not be viewed as a rigid and costly prerequisite but as a flexible, low-overhead component of the CoSTA* framework that can be dynamically and semi-automatically constructed.

## S Stability under Random Seeds

**Protocol.** We randomly selected $N = 30$ representative tasks from the full benchmark. To probe stochastic variation we executed the CoSTA * pipeline under five independent random seeds $s_1, \ldots, s_5$, keeping the data, prompts and evaluation code fixed and $\alpha = 1$ for all cases. This produced $30 \times 5 = 150$ task–seed evaluations.

**Metric.** For every seed we recorded the *mean task accuracy* $\hat{a}_s$. Let $\mu = \frac{1}{5} \sum_{s=1}^{5} \hat{a}_s$ and $\sigma = \sqrt{\frac{1}{4} \sum_{s=1}^{5} (\hat{a}_s - \mu)^2}$. We summarise run-to-run variability with the *coefficient of variation*

$$\text{CV} = \frac{\sigma}{\mu} \times 100\% = \mathbf{0.43}\%.$$

**Interpretation.** This CV well below $1\%$ signifies that random-seed stochasticity changes the *aggregate* accuracy by less than one-half of one percent. Because each extra seed incurs another 150 high-cost task executions, we judged the current slice (150 runs) to balance computational budget and statistical precision.

# T    RATIONALE FOR NORMALIZING BENCHMARK ACCURACY SCORES

A key aspect of our methodology is the normalization of benchmark accuracy scores for each tool within its specific subtask category. This normalization is critical for the effective functioning of the $A^*$ search algorithm when finding an optimal, cost-sensitive toolpath. Here, we elaborate on the reasoning behind this design choice.

Our goal is to select the best available tool for each subtask that composes the optimal toolpath. The primary reason for normalizing benchmark scores is to enable a **fair comparison** between tools that are evaluated on different subtasks using different performance metrics (e.g., mAP for object detection, CLIP score for image similarity) with vastly different natural scales. This is critical as we need to compare different toolpaths, each composed of a sequence of tool calls.

For instance, a top-performing object detection model like Grounding DINO might achieve a mean Average Precision (mAP) of 0.6, while a standard image editing model achieves a CLIP score of 0.95. If not normalized, the $A^*$ search would unfairly favor the tool with the numerically higher score, even if the 0.6 mAP represents a far superior *relative* performance for its specific task.

Consider this example:

- **Path 1** uses a sequence of top-tier models for detection and recoloring, with benchmark scores of 0.6 (mAP) and 0.96 (CLIP).
- **Path 2** uses a single model that performs the task directly with a score of 0.95 (CLIP).

Without normalization, the agent would incorrectly view Path 2 as being of higher quality than Path 1. By normalizing tools' metrics for each subtask, we ensure the best tool for a given job is always ranked highly (i.e., its score approaches 1.0). This allows the $A^*$ search to make a meaningful comparison between diverse toolpaths, preventing the arbitrary scales of different metrics from biasing its decisions. If we did not normalize the values, any path involving YOLO or Grounding DINO would likely never be selected over a path without them, even if the former path is capable of generating superior outputs.

A few other reasons for normalization include:

- **Makes Relative Performance Gaps Explicit:** Normalization highlights the relative drop in quality between competing tools for a subtask. A small absolute difference between two tools' raw scores (e.g., 0.20 vs. 0.25) can represent a significant performance gap (20% relative difference). Normalization ensures this relative shortfall is properly weighted in the agent's heuristic.
- **Compatibility with Heuristic Formula:** Our heuristic formula, which incorporates a $(2 - \text{Quality})$ term, is designed to operate on values within the $[0, 1]$ range. Normalization is therefore a technical necessity to ensure the mathematical stability and correctness of the heuristic calculation.

# U    QUALITATIVE SCENARIOS OF COSTA'S ADVANTAGES OVER GENERATIVE MODELS

While quantitative metrics demonstrate the superior performance of CoSTA, a qualitative analysis reveals common scenarios where our method's structured, tool-based approach outperforms end-

to-end solutions like Gemini and GPT-4o. These scenarios highlight the benefits of explicit task decomposition and specialized tool use.

Some common situations where our method shows a distinct advantage include:

- **Complex Multi-Turn Instructions:** For prompts with three or more sequential edits, large generative models often miss steps, leaving the editing incomplete. COSTA's structured decomposition explicitly handles each instruction as a distinct subtask, ensuring high reliability and completeness.

- **Logical Sequences:** GPT/Gemini often fail to devise a correct logical editing order where one subtask builds upon another (e.g., replacing an object before recoloring it). This can significantly reduce output quality or cause subtask failures. The intelligent planner in COSTA* correctly identifies these dependencies and devises a logical ordering of subtasks.

- **Text-in-Image Editing:** COSTA* performs precise text manipulation while preserving stylistic elements and background details by using specialized tools for text detection, removal, and rewriting. In contrast, GPT/Gemini often struggle with this, failing to maintain visual and textual consistency and sometimes introducing artifacts.

- **Realistic Object Replacement:** When replacing objects, generative models can sometimes generate items with unrealistic sizes, lighting, or positions that appear unnatural. COSTA* uses a more controlled process that often leads to more contextually appropriate and realistic replacements.

- **Integrity in Recoloration:** COSTA* is designed to preserve an object's original contents, texture, and design during recoloring tasks. Generative models can fail on complex objects, altering their shape or texture, as demonstrated in our qualitative comparisons in Figure 12.

- **Context Preservation:** COSTA* excels at preserving the overall image context by modifying only the specified elements. In contrast, generative models may misunderstand prompts and introduce unwanted artifacts or alter unrelated parts of the image, as seen in the third example of Figure 12.

## REPRODUCIBILITY STATEMENT

We are committed to ensuring the full reproducibility of our research. The complete source code for the COSTA* agent is provided as supplementary material, accessible at the following anonymous repository: https://anonymous.4open.science/r/CoSTAR-653A. This repository includes a detailed README.md file containing instructions for environment setup, dependency installation, and step-by-step guidance for running the experiments presented in the paper. We also provide a demo notebook for quick visualization and testing of our pipeline. All key architectural and algorithmic details are described in Section 4. Our experimental setup, evaluation metrics, and the baselines used for comparison are detailed in Section 5. The construction and composition of our novel benchmark dataset are thoroughly documented in Appendix H. Furthermore, the core data structures required by our method, including the Model Description Table (MDT) and the Benchmark Table (BT), are provided in full in Appendix K and Appendix L, respectively. Our human evaluation protocol is outlined in Appendix F to ensure transparency in our assessment process.

## V  USE OF LARGE LANGUAGE MODELS

In compliance with the conference guidelines, we disclose the use of Large Language Models (LLMs) in two distinct capacities for this work: (1) as a core architectural component of our proposed COSTA agent, and (2) as a general-purpose tool for assisting with manuscript preparation.

**LLM as a Core Research Component.**  An LLM is a fundamental part of the COSTA framework, where it functions as the high-level planner. As detailed throughout the paper, particularly in Section 4, the LLM's primary responsibility is to decompose complex, multi-turn image editing instructions into a structured subtask tree. This decomposition intelligently prunes the vast search space, enabling the subsequent low-level $A^*$ search to find an optimal toolpath efficiently. The specific model employed for this planning task within our experiments was GPT-4o. This use is integral to our research contribution and is described extensively in the main body of the paper.

**LLM as a Writing and Assisting Tool.**   We also utilized Google's Gemini as a general-purpose assistant for writing and document preparation. Its role included refining language for clarity and flow, checking for grammatical consistency, and rephrasing sentences. For all such uses, the human authors directed the content generation, critically reviewed all outputs for accuracy, and edited the text to ensure it faithfully represents our work.

## W ALGORITHMS

---

**Algorithm 1:** A* Search for Optimal Toolpath

---

**Input:** Tool Subgraph $G_{ts}$, Benchmark Table $BT$, Tradeoff Parameter $\alpha$, Quality Threshold

**Output:** Optimal Execution Path

**Step 1: Initialize Search**

Initialize Priority Queue $Q$;

Initialize $g(x) \leftarrow \infty$ for all nodes except root;

Precompute heuristic values for all nodes: **foreach** $v$ *in* $G_{ts}$ **do**
    $h(v) \leftarrow$ `CalculateHeuristic(`*BT, v, $\alpha$*`)`;

Initialize Start Node: Set Input Image as Root Node $r$;

$g(r) \leftarrow 0$;

$f(r) \leftarrow h(r)$;

Push $(f(r), [r])$ into $Q$;

Mark $r$ as Open;

**while** *$Q$ is not empty* **do**
    $(f(x), \text{current\_path}) \leftarrow \text{Pop}(Q)$;
    $x \leftarrow \text{LastNode}(\text{current\_path})$;
    **if** *$x$ is a leaf node* **then**
       **return** *current\_path*
    **foreach** *neighbor $y$ in* `Neighbors(`$x$`)` **do**
       $c(y) \leftarrow$ `CalculateActualCost(`$y$`)`;
       $q(y) \leftarrow$ `CalculateActualQuality(`$y$`)`;
       $g(y)_{new} \leftarrow$ `ComputeExecutionCost(`$g(x), c(y), q(y), \alpha$`)`;
       **if** `QualityCheck(`$y$`)` $\geq$ *Quality Threshold* **then**
          $g(y) \leftarrow$ `Min(`$g(y)_{new}, g(y)$`)`;
       **else**
          $g(y)_{new2} \leftarrow$ `RetryMechanism(`$y$`)`;
          **if** `QualityCheck(`$y$`)` $\geq$ *Quality Threshold* **then**
             $g(y)_{final} \leftarrow g(y)_{new} + g(y)_{new2}$;
             $g(y) \leftarrow$ `Min(`$g(y)_{final}, g(y)$`)`;
          **else**
             **continue**; Node remains unexplored
       $f(y) \leftarrow g(y) + h(y)$;
       Push $(f(y), \text{current\_path} + [y])$ into $Q$;

**Step 2: Output Optimal Path**

Terminate when the lowest-cost valid path is found;

**return** *Optimal Path*;

---

---

**Algorithm 2:** Tool Subgraph Construction

---

**Input:** Image $x$, Prompt $u$, Tool Dependency Graph $G_{td}$, Model Description Table $MDT$,
      Supported Subtasks $\mathcal{S}$

**Output:** Tool Subgraph $G_{ts}$

**Step 1: Generate Subtask Tree**

$G_{ss} \leftarrow$ `GenerateSubtaskTree(`*LLM, x, u, $\mathcal{S}$*`)`;

**Step 2: Build Tool Subgraph (TG)**

Initialize $G_{ts}$;

**foreach** *subtask $s_i \in V_{ss}$* **do**
    $T_i \leftarrow$ `GetModelsForSubtask(`$MDT, s_i$`)`;
    $G_{ti} \leftarrow$ `BacktrackDependencies(`$G_{td}, T_i$`)`;
    Replace $s_i$ in $G_{ss}$ with $G_{ti}$ to construct $G_{ts}$;

**return** $G_{ts}$;

---

# X  LLM PROMPT FOR GENERATING SUBTASK TREE

**You are an advanced reasoning model responsible for decomposing a given image editing task into a structured subtask tree. Your task is to generate a well-formed subtask tree that logically organizes all necessary steps to fulfill the given user prompt. Below are key guidelines and expectations:**

## X.1  UNDERSTANDING THE SUBTASK TREE

A subtask tree is a structured representation of how the given image editing task should be broken down into smaller, logically ordered subtasks. Each node in the tree represents an atomic operation that must be performed on the image. The tree ensures that all necessary operations are logically ordered, meaning a subtask that depends on another must appear after its dependency.

## X.2  STEPS TO GENERATE THE SUBTASK TREE

1. **Step 1:** Identify all relevant subtasks needed to fulfill the given prompt.

2. **Step 2:** Ensure that each subtask is logically ordered, meaning operations dependent on another should be placed later in the path.

3. **Step 3:** Each subtask should be uniquely labeled based on the object it applies to and follow the format (Obj1 → Obj2) where Obj1 is replaced with Obj2. In case of recoloring, use (Obj → new color), while for removal, simply include (Obj) as the object being removed.

4. **Step 4:** A tree may involve multiple correct paths where subtasks are independent of each other. In such cases, a subtask may appear twice in different parts of the tree. Number such occurrences distinctly, e.g., `Subtask1(1)`, `Subtask1(2)`, ensuring clarity.

5. **Step 5:** Some tasks may have multiple valid approaches. For example, replacing a cat with a pink dog can be done in two ways:

   - `Object Replacement (Cat → Pink Dog)`
   - `Object Replacement (Cat → Dog)` → `Object Recoloration (Dog → Pink Dog)`

## X.3  LOGICAL CONSTRAINTS & DEPENDENCIES

- Ensure proper ordering, e.g., if an object is replaced and then segmented, segmentation must follow replacement.

- Operations should be structured logically so that every subtask builds upon the previous one.

## X.4  SUPPORTED SUBTASKS

Below is the complete list of available subtasks: Object Detection, Object Segmentation, Object Addition, Object Removal, Background Removal, Landmark Detection, Object Replacement, Image Upscaling, Image Captioning, Changing Scenery, Object Recoloration, Outpainting, Depth Estimation, Image Deblurring, Text Extraction, Text Replacement, Text Removal, Text Addition, Text Redaction, Question Answering Based on Text, Keyword Highlighting, Sentiment Analysis, Caption Consistency Check, Text Detection
**You must strictly use only these subtasks when constructing the tree.**

## X.5  EXPECTED OUTPUT FORMAT

The model should output the subtask tree in structured JSON format, where each node contains:

- **Subtask Name** (with object label if applicable)

- **Parent Node** (Parent subtask from which it depends)

- **Execution Order** (Logical flow of tasks)

### X.6 EXAMPLE INPUTS & EXPECTED OUTPUTS

#### X.6.1 EXAMPLE 1

**Input Prompt:** *"Detect the pedestrians, remove the car and replacement the cat with rabbit and recolor the dog to pink."*

**Expected Subtask Tree:**

```
"task": "Detect the pedestrians, remove the car and replacement
 the cat with rabbit and recolor the dog to pink",
"subtask_tree": [
        {
              "subtask": "Object Detection (Pedestrian)(1)",
              "parent": []
        },
        {
              "subtask": "Object Removal (Car)(2)",
              "parent": ["Object Detection (Pedestrian)(1)"]
        },
        {
              "subtask": "Object Replacement (Cat -> Rabbit)(3)",
              "parent": ["Object Removal (Car)(2)"]
        },
        {
              "subtask": "Object Replacement (Cat -> Rabbit)(4)",
              "parent": ["Object Detection (Pedestrian)(1)"]
        },
        {
              "subtask": "Object Removal (Car)(5)",
              "parent": ["Object Replacement (Cat -> Rabbit)(4)"]
        },
        {
              "subtask": "Object Recoloration (Dog ->
                                            Pink Dog)(6)",
              "parent": ["Object Replacement (Cat -> Rabbit)(3)",
                          "Object Removal (Car)(5)"]
        }
    ]
```

#### X.6.2 EXAMPLE 2

**Input Prompt:** *"Update the closed signage to open while detecting the trash can and pedestrian crossing for better scene understanding. Also, remove the people for clarity."*

**Expected Subtask Tree:**

```
"task": "Update the closed signage to open while detecting the
 trash can and pedestrian crossing for better scene
 understanding. Also, remove the people for clarity.",

"subtask_tree": [
    {
        "subtask": "Text Replacement (CLOSED -> OPEN)(1)",
        "parent": []
    },
    {
        "subtask": "Object Detection (Pedestrian Crossing)(2)",
        "parent": ["Text Replacement (CLOSED -> OPEN)(1)"]
    },
```

```
    {
        "subtask": "Object Detection (Trash Can)(3)",
        "parent": ["Text Replacement (CLOSED -> OPEN)(1)"]
    },
    {
        "subtask": "Object Detection (Pedestrian Crossing)(4)",
        "parent": ["Object Detection (Trash Can)(3)"]
    },
    {
        "subtask": "Object Detection (Trash Can)(5)",
        "parent": ["Object Detection (Pedestrian Crossing)(2)"]
    },
    {
        "subtask": "Object Removal (People)(6)",
        "parent": ["Object Detection (Pedestrian Crossing)(4)",
                    "Object Detection (Trash Can)(5)"]
    }
]
```

## X.7   FINAL TASK

**Now, using the given input image and prompt, generate a well-structured subtask tree that adheres to the principles outlined above.**

- Ensure logical ordering and clear dependencies.
- Label subtasks by object name where needed.
- Structure the output as a JSON-formatted subtask tree.

**Input Details:**

- Image: `input_image`
- Prompt: `User Prompt`
- Supported Subtasks: (See the list above)

**Now, generate the correct subtask tree. Before you generate the tree, ensure that for every possible path, all required subtasks are included and none are skipped.**

# Y    LLM PROMPT FOR GETTING BOUNDING BOX AND TEXT FOR REPLACEMENT

**You are given an image containing text, where each word has associated bounding box coordinates. The existing text and their corresponding bounding boxes are as follows:**

- **"THIS"**: (281,438,502,438,502,494,281,494)
- **"IS"**: (533,437,649,440,647,497,531,493)
- **"A"**: (667,444,734,444,734,492,667,492)
- **"NICE"**: (214,504,810,502,811,649,214,651)
- **"STREET"**: (68,674,915,640,924,859,77,893)

The user wants to replace this text with:



**"THIS IS NOT A NICE STREET"**



## Y.1    YOUR TASK

You must determine which words in the image should be removed and which words need to be rewritten to ensure a smooth transition to the new text. The goal is to maintain spatial coherence while ensuring that the updated text fits naturally within the image.

## Y.2    GUIDELINES FOR TEXT REPLACEMENT

1. **Identify Words to Remove:**
   - Any word that needs to be replaced or modified should be marked for removal.
   - If the new text introduces an additional word, the surrounding words should also be removed and rewritten to maintain proper spacing.

2. **Determine Placement for New Words:**
   - If a word or phrase is being replaced (e.g., `"GOOD BOY"` → `"BAD GIRL"`), use a single bounding box that covers the area of both words instead of providing separate locations.
   - If new words need to be inserted, ensure that adjacent words are also rewritten to provide sufficient space for readability.
   - If the new text is longer than the original, adjust placements accordingly:
     - Remove and rewrite words from the next or previous line if needed.
     - If necessary, split the updated text into two separate lines and provide distinct bounding boxes for each.

3. **Bounding Box Adjustments:**
   - If text placement changes, the bounding box should be expanded or shifted to accommodate the new words.
   - Ensure that all bounding boxes align with the natural flow of the text in the image.

## Y.3    EXAMPLE CASE FOR CLARITY

**Input Scenario:**
*Original Text:* "I AM A GOOD BOY" *Replacement Text:* "I AM A BAD GIRL"
**Expected Output:**

- Remove: `"GOOD"` and `"BOY"`
- Write: `"BAD GIRL"`
- Bounding Box for "BAD GIRL": (Bounding box covering the area where "GOOD BOY" was originally written)

If `"BAD GIRL"` doesn't fit naturally within the same space, adjust the bounding box or split it into multiple lines.

