# OpenReview forum: "CoSTA*: Cost-Sensitive Toolpath Agent for Multi-turn Image Editing"
_ICLR.cc/2026/Conference — Submitted to ICLR 2026_

### Official Review · Reviewer_BKHU · 2025-10-20

**Soundness:** 2
**Presentation:** 2
**Contribution:** 2
**Rating:** 4
**Confidence:** 4

**Summary:**

This paper presents a method called CoSTA* to improve multi-step image editing. The main idea is to break down an editing task into a sequence of smaller subtasks and find the most efficient path of AI tools to complete them. The paper notes that LLMs are good at planning subtasks but are not accurate at estimating the cost or quality of the tools needed for each step. CoSTA* combines the planning strength of LLMs with a search algorithm. First, an LLM creates a plan, which helps to narrow down the number of possible tools. Then, an A* search algorithm finds the best path through these tools, balancing both cost and quality. VLMs are used to check the output. If a tool fails, the system updates its knowledge about that tool's cost and quality, allowing the search to find an alternative path. To test their method, the authors created a new benchmark of image editing tasks and showed that CoSTA* performs better in both cost and quality.

**Strengths:**

1. The paper is easy to understand.
2. The motivation is clear.
3. The engineering efforts in designing all the editing tools are appreciated and may be helpful to the community.

**Weaknesses:**

My major concerns are two-fold, focusing on its motivation and technical novelty.

1. The primary motivation of optimizing for cost, defined as inference time, seems insufficiently justified for the domain of multi-turn image editing. This type of task is often performed asynchronously; users typically submit a job and retrieve the results later, making real-time performance a secondary concern. Furthermore, this focus on model inference time overlooks two key factors: 1) Editing tools are rapidly evolving, meaning any cost calculation based on current inference times is ephemeral. 2) Real-world applications often rely on APIs, which introduce network latency and queuing delays that are unrelated to model inference and can dominate the total wait time.

2. The paper presents limited technical novelty, as its core components are well-established concepts within the agent and planning communities. Decomposing tasks into trees or graphs of dependencies is a well-studied paradigm, seen in prior work such as VISPROG, ViperGPT, HuggingGPT, Tree of Thoughts, and Graph of Thoughts. Using VLMs to evaluate intermediate steps and guide the subsequent action is a common strategy in building autonomous agentic workflows. The necessity of an A* search is questionable. A simpler, rough plan executed with a ReAct-style pattern (e.g., HuggingGPT + ReAct) is likely sufficient. The success of any plan ultimately hinges on the quality of the underlying tools. If a tool cannot produce the desired result, no amount of sophisticated planning or searching can salvage the outcome. The true bottleneck is often tool capability, not the planning algorithm.

**Questions:**

While I am not an expert in designing image editing benchmarks, I have two comments:

1. A potential concern is that the benchmark may be implicitly tailored to the specific editing tools and patterns developed by the authors. This could create an unfair comparison against general-purpose methods that have not been specifically optimized for this task suite.

2. The diversity and sample number might not be sufficient to draw robust and generalizable conclusions.

---

> ### Author Response · Authors · 2025-11-24
> **Response to Reviewer BKHU (1 of 2)**
>
> Thank you for your detailed feedback! We address your comments below.
>
> > Q1: The motivation of optimizing for cost (inference time) seems unjustified; real-time performance is secondary.
>
> The inference latency is indeed a critical bottleneck because state-of-the-art generation requires computationally heavy models, and agentic workflows multiply this cost through sequential tool chains.
> Even in asynchronous settings, reducing total execution time is decisive for user retention and economic viability. Significant research efforts have been dedicated to this problem (e.g., **"Beta Sampling is All You Need" [Lee et al., WACV 2025]**, **"NAMI" [Ma et al., arXiv 2025]**), implying that latency is a primary concern. CoSTA* addresses this by LLM + A* hierarchical planning to mitigate the "agentic multiplier" effect.
>
> > Q2: Focus on model inference time overlooks two key factors: 1) Editing tools are rapidly evolving. 2) Real-world applications often rely on APIs, which introduce network latency.
>
> CoSTA* is a general agentic framework that can be adapted to evolved tools and API costs. There changes are captured by exploration and the updated model decription/benchmark tables. It improves the efficiency by optimizing path length and tool choices on the path, which does not depend on specific choices of tools or APIs.
> *   **Ephemeral Costs:** CoSTA*'s exploration can adapt to changes in the tool costs and find paths based on the evolved tools. The dynamic **Benchmark Table** (Appendix Q) automatically updates heuristics as tools evolve.
>
> *   **API Latency:** While network latency is an unavoidable overhead for all methods, CoSTA* focuses on optimizing the controllable components: model execution time and path length. By finding more efficient toolpaths with fewer steps and retrials, CoSTA* inherently reduces the total accumulated network overhead, resulting in a **~20% reduction** in overall execution cost (Table 9).
>
> > Q3: The paper presents limited technical novelty. The necessity of an A* search is questionable. The true bottleneck is often tool capability, not the planning algorithm.
>
> The success requires **both** capable tools and effective planning. CoSTA* combines the complementary strengths of ReAct LLM agent and novel cost-sensitive A* search. Our experimental results show that ReAct methods fail on complex tasks that CoSTA* excel on.
>
> 1.  ReAct vs. A* Search: Baselines like **GenArtist** and **CLOVA** adopt the ReAct paradigm. In contrast to CoSTA*, they struggle on complex, multi-turn tasks (0.61 accuracy vs. CoSTA*'s 0.95 on tasks with 7-8 subtasks, as shown in Table 3) because greedy planning cannot foresee dead ends. CoSTA*'s A* search enables **backtracking** and **global optimization** of tool paths (Figure 14).
> 2.  **Tool Capability & Redundancy:** Since individual tools are imperfect, our framework incorporates **tool redundancy** (multiple tools for similar subtasks). But managing this redundancy introduces combinatorial complexity that makes simple search infeasible and greedy planning suboptimal. CoSTA* is specifically designed to navigate this space efficiently.
> 3.  **Empirical Evidence:** The **"Fair Comparison"** ablation (detailed in Q4 below) restricts CoSTA* to **exactly the same tools** as the baselines yet still yields a **~41% overall improvement**. This demonstrates that CoSTA* planning's ability to recover from errors is critical and it distincts from tool capability.

---

> > ### Author Response · Authors · 2025-11-24
> > **Response to Reviewer BKHU (2 of 2)**
> >
> > > Q4: A potential concern is that the benchmark may be implicitly tailored to the specific editing tools and patterns developed by the authors.
> >
> > We addressed this concern in **Appendix D** by conducting a rigorous set of "Fair Comparison" ablation studies. In these experiments, we stripped CoSTA* of its specialized tool advantage and restricted it to use **only** restricted tools and subtasks adopted by the baselines. This ensures that any performance difference is attributable solely to CoSTA*'s planning vs. the baselines' orchestration methods.
> >
> > **Table 10: Fair comparison with VisProg/CLOVA using their restricted toolset**
> > | Subtasks | CoSTA* | VisProg | CLOVA | Diff w/ CoSTA* |
> > | :--- | :--- | :--- | :--- | :--- |
> > | 1-2 Subtasks | 0.510 | 0.498 | 0.504 | -1.7% |
> > | 3-4 Subtasks | 0.551 | 0.489 | 0.496 | -10.6% |
> > | 5-6 Subtasks | 0.573 | 0.446 | 0.451 | -21.7% |
> > | 7-8 Subtasks | 0.460 | 0.301 | 0.310 | **-33.5%** |
> > | **Overall** | **0.525** | 0.436 | 0.446 | **-16.0%** |
> >
> > **Table 11: Fair comparison with GenArtist using GenArtist's toolset**
> > | Subtasks | CoSTA* | GenArtist | Diff w/ CoSTA* |
> > | :--- | :--- | :--- | :--- |
> > | 1-2 Subtasks | 0.508 | 0.506 | -0.39% |
> > | 3-4 Subtasks | 0.578 | 0.548 | -5.21% |
> > | 5-6 Subtasks | 0.606 | 0.503 | -17.11% |
> > | 7-8 Subtasks | 0.515 | 0.392 | **-23.89%** |
> > | **Overall** | **0.553** | 0.495 | **-10.55%** |
> >
> > **Table 12: Fair comparison of CoSTA∗ with all baselines using a restricted subtask list common to all while using our own toolset for these corresponding allowed subtasks. Difference with CoSTA∗ is calculated as the average of all other baselines vs. CoSTA∗.**
> > | Subtasks | CoSTA* | VisProg | CLOVA | InstructPix2Pix | MagicBrush | GenArtist | Diff w/ CoSTA* |
> > | :--- | :--- | :--- | :--- | :--- | :--- | :--- | :--- |
> > | 1-2 Subtasks | 0.550 | 0.498 | 0.504 | 0.497 | 0.501 | 0.506 | -8.87% |
> > | 3-4 Subtasks | 0.632 | 0.489 | 0.496 | 0.477 | 0.503 | 0.548 | -20.75% |
> > | 5-6 Subtasks | 0.654 | 0.446 | 0.451 | 0.390 | 0.402 | 0.503 | -32.96% |
> > | 7-8 Subtasks | 0.599 | 0.301 | 0.310 | 0.274 | 0.315 | 0.392 | **-46.85%** |
> > | **Overall** | **0.610** | 0.436 | 0.446 | 0.413 | 0.434 | 0.495 | **-27.08%** |
> >
> > These results demonstrate that even when "handicapped" to the same tools as general-purpose baselines, CoSTA* achieves significantly higher accuracy, particularly on complex tasks (**+46.85% relative improvement on 7-8 subtasks**).
> >
> >
> > > Q5: The diversity and sample number might not be sufficient to draw robust and generalizable conclusions.
> >
> > Our dataset covers **550 distinct edits** (1-8 turns per prompt), which ensures the diversity and the complexity of tasks. The generalizability of our conclusions is validated by evaluating CoSTA* on large-scale public datasets, on which CoSTA* outperforms all the baselines.
> >
> > 1.  **Internal Diversity:** Our dataset is designed to fill a gap in task complexity; unlike previous benchmarks capped at 1-2 turns, ours tests coherence over long horizons (up to 8 subtasks) and covers diverse multimodal logics (Figures 6 & 10).
> > 2.  **External Validation:** To rule out dataset bias, we evaluated CoSTA* on **MagicBrush** and **EMU-Edit** (Appendix A). CoSTA* consistently outperforms baselines (e.g., **0.95 accuracy vs. 0.81** on EMU-Edit, Table 8), confirming that our conclusions are robust and generalizable.

---

> > > ### Author Response · Authors · 2025-11-27
> > > **Follow up on the response**
> > >
> > > Dear Reviewer,
> > >
> > >
> > > Thank you again for your time and helpful suggestions. We wanted to follow up to confirm that our responses have fully resolved your concerns. We remain available and are happy to provide any further clarifications should you have additional questions during the remaining discussion period.
> > >
> > >
> > > Sincerely,
> > >
> > > The authors.

---

### Official Review · Reviewer_Gbp4 · 2025-10-30

**Soundness:** 3
**Presentation:** 3
**Contribution:** 3
**Rating:** 6
**Confidence:** 4

**Summary:**

This paper presents CoSTA*, a hierarchical agent framework for complex, multi-turn image editing tasks that combines LLM-based high-level planning with cost-sensitive A* search for low-level tool selection. The system addresses the challenge of efficiently orchestrating sequences of editing operations (toolpaths) while balancing quality and computational cost. CoSTA* operates in three stages: (1) an LLM decomposes the editing task into a subtask tree, (2) this tree prunes a Tool Dependency Graph (TDG) to create a focused subgraph, and (3) A* search finds the optimal toolpath using a cost function f(x) = g(x) + h(x) that combines actual execution cost with heuristic estimates, controlled by a tradeoff parameter α. The system incorporates real-time VLM-based quality checking with adaptive retry mechanisms. The authors construct a benchmark of 121 curated image-editing tasks with 1-8 subtasks (550 total edits) covering both image-only and text-in-image operations. Experiments demonstrate CoSTA* achieves 94% accuracy overall, substantially outperforming existing agents (GenArtist: 73%, VisProg: 62%) and closed-source models (Gemini 2.0: 81%, GPT-4o: 78%), while offering explicit control over cost-quality tradeoffs through the α parameter.

**Strengths:**

1、 Multi-turn image editing with cost control is a genuine challenge for current systems. The motivation for combining symbolic search with neural planning is compelling and addresses real limitations of pure LLM-based agents.

2、The hierarchical combination of LLM subtask decomposition with A* search on a pruned tool graph is innovative. This design elegantly leverages LLM strengths (high-level reasoning) while mitigating weaknesses (poor cost/quality estimation) through structured search.

3、The analysis showing LLM pruning reduces search space from 100,000+ nodes to 15-20 nodes demonstrates practical scalability of the approach.

**Weaknesses:**

1、While the integration is novel, the core techniques—LLM task decomposition, A* search, VLM evaluation—are standard. The A* formulation itself is relatively straightforward, and the heuristic h(x) is computed via simple recursive propagation from empirical benchmarks rather than learned

2、All experiments use a single A100 GPU. How does the approach scale to larger tool libraries (50+ tools), deeper task hierarchies (10+ subtasks), or real-time editing scenarios? The paper provides no runtime analysis or computational complexity discussion beyond node count reduction.

**Questions:**

see Weaknesses

---

> ### Author Response · Authors · 2025-11-24
> **Response to Reviewer Gbp4**
>
> Thank you for your detailed feedback! We address your comments below.
>
> > Q1: The core techniques are standard. The A* formulation is relatively straightforward, and the heuristic is computed via simple propagation rather than learned.
>
> Our technical novelty lies on the **hierarchical, cost-sensitive planning** to address challenging multi-turn image editing tasks. It is non-trivial as the planning scheme combines the complementary strengths of LLM planning and A* search to overcome their individual limitations on those tasks.
>
> 1.  **Synergy over Isolation:** LLMs provide semantic-level reasoning but lack numerical precision to perform quality-cost trade-off, while A* search ensures optimality but is computationally intractable for the vast tool search space of multi-turn editing (the global tool graph). Our hierarchical planning leverages LLMs to prune the graph so A* only needs to search on much smaller and sparser subgraphs, achieving results neither can alone.
> 2.  **Why Empirical Heuristics:** In contrast to costly training on massive data, we chose empirical propagation over learned heuristics to achieve better **data efficiency** and **interpretability**----the $\alpha$ parameter grants users delicate control over the cost-quality trade-off, which is hardly achieved by training heuristics.
>
>
>
> > Q2: How does the approach scale to larger tool libraries (50+ tools) and deeper task hierarchies (10+ subtasks)? No runtime analysis or computational complexity discussion beyond node count reduction.
>
> CoSTA* scales efficiently to larger tool libraries because the LLM acts as a high-level pruner, reducing the effective branching factor of A* search to only "relevant tools," rendering the search speed independent of global library size.
>
> 1.  **Computational Complexity:** Standard A* is $O(b^d)$, where $b$ is the branching factor and $d$ is the depth. Without pruning, $b$ equals the total number of possible subtasks (intractable). CoSTA* uses the LLM to identify the specific sequence of required subtasks (e.g., "Object Removal"), and the Model Description Table (MDT) populates only relevant tools. This reduces $b$ to typically 2-3 (the number of tools per subtask), making the complexity tractable at $O(3^d)$ and keeping the search graph small (~15-20 nodes) even if the library grows to 1000+ tools.
> 2.  **Task Depth (10+ Subtasks):** We validated our planner on **15 additional samples with 10-15 subtasks**, all of which were successful. Unlike LLM planners that suffer from context drift on long-horizon tasks, local A* search is **deterministic**, ensuring the optimization logic remains rigorous regardless of depth.
> 3.  **Runtime Analysis:** We provide execution cost analysis in **Table 9**. The "planning overhead" caused by A* search is negligible (in milliseconds) when compared to the inference time of the generative models (often >10 seconds per call). By finding "cheaper" toolpaths (e.g., using a faster specialist model to replace an expensive generalist model), CoSTA* often achieves a **lower** total execution time (~53s vs. ~67s of baselines) despite the search step.

---

> > ### Author Response · Authors · 2025-11-27
> > **Follow up on the response**
> >
> > Dear Reviewer,
> >
> >
> > Thank you again for your time and helpful suggestions. We wanted to follow up to confirm that our responses have fully resolved your concerns. We remain available and are happy to provide any further clarifications should you have additional questions during the remaining discussion period.
> >
> >
> > Sincerely,
> >
> > The authors.

---

### Official Review · Reviewer_zfMS · 2025-10-31

**Soundness:** 3
**Presentation:** 3
**Contribution:** 3
**Rating:** 6
**Confidence:** 3

**Summary:**

The paper proposes a novel approach, CoSTA∗, for multi-turn image editing, combining large language models (LLMs) and A∗ search to efficiently plan and execute tool paths. It targets improving the cost and quality trade-offs in complex image editing tasks, specifically addressing multi-step workflows. The authors introduce an agentic mechanism that decomposes a complex task into smaller subtasks, which are then optimized using a hierarchical planning system. CoSTA∗ outperforms existing state-of-the-art image editing models in terms of both cost efficiency and output quality, offering a dynamic trade-off mechanism that adjusts based on user-defined preferences.

**Strengths:**

1. The integration of LLMs for task decomposition with A∗ search for tool path optimization is a novel and promising approach, addressing key challenges in multi-turn image editing.

2. The dynamic tuning of cost and quality through the use of a tunable coefficient (α) provides users with flexibility and optimization, offering Pareto-optimal solutions for both performance metrics.

3. The paper introduces a new, challenging benchmark for multi-turn image editing tasks, which contributes significantly to the field by allowing more rigorous evaluation of editing agents.

**Weaknesses:**

1. The system relies on prior knowledge from benchmark datasets, which may not always be available or applicable for all types of tasks.

2. Although CoSTA∗ excels in multimodal tasks, it may still face challenges when adapting to new, previously unseen tools or modalities. The adaptability in real-world usage needs to be tested more thoroughly outside of controlled benchmarks.

3. The method shows strong performance in the experimental setting, but its scalability to larger, more complex workflows could be an area for further investigation, especially in real-world applications.

**Questions:**

see my weaknesses.

---

> ### Author Response · Authors · 2025-11-24
> **Response to Reviewer zfMS**
>
> Thank you for your detailed feedback! We address your comments below.
>
> > Q1: The system relies on prior knowledge from benchmark datasets, which may not always be available or applicable for all types of tasks.
>
> The "prior knowledge" required minimal efforts to collect, primarily execution time and a relative quality score. While a pre-computed table optimizes initial performance, CoSTA* is designed to function without it via the **"Cold Start" mechanism (detailed in Appendix Q)**.
> *   **Dynamic benchmark table:** The system can initialize with a placeholder. As it executes tasks, it records real-time performance (via the VLM feedback loop), dynamically populating and updating the Benchmark Table.
> *   **Self-Correction:** This allows the agent to "learn" the effective costs and quality of tools on-the-fly during inference, effectively eliminating dependency on offline datasets.
>
> > Q2: Although CoSTA∗ excels in multimodal tasks, it may still face challenges when adapting to new, previously unseen tools or modalities.
>
> 1.  **Adaptability:** CoSTA* is a training-free, cost-sensitive planning framework of inference. Unlike policy-based agents that "memorize" specific tool interactions during training, CoSTA* is inherently modular. Adapting to a completely new tool or modality only requires adding a single row to the Model Description Table. The planner (LLM) and search algorithm (A*) directly adapt to the new capability without any retraining or fine-tuning.
> 2.  **Generalizability:** To demonstrate our results are not artifacts of a specific dataset, we evaluated CoSTA* on standard public benchmarks, **MagicBrush** and **EMU-Edit**. As shown in **Tables 7 and 8 (Appendix A)**, CoSTA* consistently outperforms baselines in these thrid-party benchmarks, verifying its robustness in diverse real-world applications.
>
> > Q3: Scalability to larger, more complex workflows could be an area for further investigation.
>
> CoSTA* scales up better than pure LLM approaches in two dimensions:
> *   **Scalable to more tools:** A key advantage brought by our hierarchical planning is that planning complexity does not explode with the size of the toolset. The **LLM-based pruning (Stage 1)** acts as a filter, narrowing down a tool graph of potentially hundreds of nodes to a much smaller, relevant subgraph (typically 15-20 nodes).
> *   **Scalable to complex tasks with more subtasks (turns):** While current SOTA agents often degrade rapidly after 2-3 turns, our experiments demonstrate CoSTA*'s stability on tasks with up to **8 sequential subtasks** (Table 3). The A* search's ability to backtrack and recover from errors enables it to navigate deep dependency chains, which typically break greedy, ReAct-style agents.

---

> > ### Author Response · Authors · 2025-11-27
> > **Follow up on the response**
> >
> > Dear Reviewer,
> >
> >
> > Thank you again for your time and helpful suggestions. We wanted to follow up to confirm that our responses have fully resolved your concerns. We remain available and are happy to provide any further clarifications should you have additional questions during the remaining discussion period.
> >
> >
> > Sincerely,
> >
> > The authors.

---

### Official Review · Reviewer_o7Vm · 2025-11-01

**Soundness:** 2
**Presentation:** 3
**Contribution:** 2
**Rating:** 4
**Confidence:** 4

**Summary:**

This paper introduces CoSTA*, a cost-sensitive toolpath agent designed for multi-turn image editing. The framework integrates a large language model (LLM) for high-level subtask planning with an A* search algorithm for low-level optimization. It constructs a Tool Dependency Graph (TDG) and a Model Description Table (MDT) to define tool relations and supported subtasks, and incorporates a cost-quality tradeoff via tunable coefficient α. The authors also propose a new benchmark for complex multi-turn editing and claim superior performance over recent systems such as GenArtist, CLOVA, MagicBrush, and InstructPix2Pix.

**Strengths:**

- The hierarchical design combining LLM-based planning and A*-based search is conceptually elegant and well-motivated. The inclusion of structured metadata (MDT and TDG) demonstrates thoughtful system engineering.
- The paper introduces a new benchmark for multi-turn multimodal editing, which may be valuable to the community if released publicly with standardized evaluation protocols.
- The idea of incorporating cost sensitivity into multi-turn image editing is interesting and practically relevant, especially for agentic workflows where computational cost varies significantly across tools.

**Weaknesses:**

- While the framework description is detailed, the technical novelty appears incremental. The “cost-sensitive A*” mainly combines standard heuristic search with ad-hoc cost-quality weighting (α). The mathematical formulation (Eq. 3–4) is not rigorously justified, and the approach reads more like system integration than algorithmic innovation.
- The proposed benchmark is not clearly standardized, and human evaluation protocols are vaguely defined. Metrics like “accuracy” and “quality” are subjective and loosely defined; CLIP-based evaluation is known to be unreliable but is still used in parts of the analysis.
- The experiments show marginal improvements over strong baselines on public datasets suggesting that gains primarily come from tool diversity rather than planning effectiveness. Ablations confirm that removing multimodality or tool dependency graphs causes large drops, implying the method’s advantage lies in data engineering rather than algorithmic design.

**Questions:**

See Weaknesses.

---

> ### Author Response · Authors · 2025-11-24
> **Response to Reviewer o7Vm**
>
> Thank you for your detailed feedback! We address your comments below.
>
> > Q1: The technical novelty appears incremental; the approach reads more like system integration than algorithmic innovation.
>
> Our core novelty is the **hierarchical, neurosymbolic planning agent** designed to address multi-turn image editing challenges and quality-cost trade-off. It combines the complementary strengths of LLM planner and tree search planner and use one to overcome the other's shortcomings.
>
> Conducting tree search is computationally infeasible in the vast space of available tools (>100,000 possible nodes). CoSTA* innovatively develops a **LLM-guided pruning**, which dynamically reduces the space to a manageable subgraph (~15-20 nodes). This structural shift makes fine-grained, cost-sensitive optimization feasible, representing a fundamental schema innovation beyond simple integration.
>
> > Q2: The “cost-sensitive A*” combines standard search with ad-hoc weighting, and the mathematical formulation (Eq. 3–4) is not rigorously justified.
>
> The heuristic formulation in Eq. (3) has been derived through rigorous empirical validation on a held-out validation set to ensure mathematical stability and effective balancing.
>
> We evaluated multiple candidates: **Cost-Only** ($h=\text{Cost}$) resulted in lower quality (0.87); **Quality-Only** ($h=2-\text{Quality}$) led to prohibitive costs (~69s); and **Linear Combinations** with different weights proved unstable (0.89 quality), often favoring low-cost/poor-quality tools. The proposed **Multiplicative** formulation achieved the best balance (0.93 quality, 58.2s cost) and is mathematically necessary to enable the explicit Pareto-optimal trade-offs shown in Figure 3.
>
> > Q3: The proposed benchmark is not clearly standardized; human evaluation and metrics are subjective/vague.
>
> Our human evaluation follows a strict, standardized protocol with a detailed rubric (0.0–1.0) detailed in **Appendix F**. It achieves high inter-rater consistency, i.e., a low variance of **0.07** across 5 human evaluators. To further aid transparency, we included 12 distinct qualitative examples in the paper (Figure 1 and Appendix) to visually assess performance. We will add more qualitative examples with corresponding accuracy scores to the updated draft to better illustrate the evaluation protocols.
>
> We rely primarily on this human evaluation for our final claims. CLIP is **only** a reward signal to guide CoSTA* within its internal feedback loop: it is not used as a performance evaluation metric.
>
> > Q4: The experiments show marginal improvements on public datasets. The method’s advantage lies in data engineering rather than algorithmic design.
>
> Improvements on previous datasets are marginal because tasks in these benchmarks are much simpler (1-2 turns at the maximum) and can be easily handled by the baselines. CoSTA*'s decisive advantage lies in **planning for complex multi-turn editing**, demonstrated by our "Fair Comparison" ablation, where CoSTA* still outperforms baselines by **~41%** using the **same restricted toolset** adopted by the baseline approaches.
>
> While specific tools are required for multimodal tasks, tool availability alone is insufficient without a planner to wield them. Our results in **Tables 10, 11, and 12** (Appendix D) confirm that without CoSTA*'s A* planner, baselines fail on complex tasks (7-8 subtasks) even when given identical tools.
>
> **Table 10: Fair comparison with VisProg/CLOVA (Restricted Toolset)**
> | Subtasks | CoSTA* | VisProg | CLOVA | Diff w/ CoSTA* |
> | :--- | :--- | :--- | :--- | :--- |
> | 1-2 Subtasks | 0.510 | 0.498 | 0.504 | -1.7% |
> | 3-4 Subtasks | 0.551 | 0.489 | 0.496 | -10.6% |
> | 5-6 Subtasks | 0.573 | 0.446 | 0.451 | -21.7% |
> | 7-8 Subtasks | 0.460 | 0.301 | 0.310 | **-33.5%** |
> | **Overall** | **0.525** | 0.436 | 0.446 | **-16.0%** |
>
> **Table 11: Fair comparison with GenArtist (Restricted Toolset)**
> | Subtasks | CoSTA* | GenArtist | Diff w/ CoSTA* |
> | :--- | :--- | :--- | :--- |
> | 1-2 Subtasks | 0.508 | 0.506 | -0.39% |
> | 3-4 Subtasks | 0.578 | 0.548 | -5.21% |
> | 5-6 Subtasks | 0.606 | 0.503 | -17.11% |
> | 7-8 Subtasks | 0.515 | 0.392 | **-23.89%** |
> | **Overall** | **0.553** | 0.495 | **-10.55%** |
>
> **Table 12: Fair comparison with All Baselines (Common Toolset)**
> | Subtasks | CoSTA* | VisProg | CLOVA | InstructPix2Pix | MagicBrush | GenArtist | Diff w/ CoSTA* |
> | :--- | :--- | :- | :--- | :--- | :--- | :--- | :--- |
> | 1-2 Subtasks | 0.550 | 0.498 | 0.504 | 0.497 | 0.501 | 0.506 | -8.87% |
> | 3-4 Subtasks | 0.632 | 0.489 | 0.496 | 0.477 | 0.503 | 0.548 | -20.75% |
> | 5-6 Subtasks | 0.654 | 0.446 | 0.451 | 0.390 | 0.402 | 0.503 | -32.96% |
> | 7-8 Subtasks | 0.599 | 0.301 | 0.310 | 0.274 | 0.315 | 0.392 | **-46.85%** |
> | **Overall** | **0.610** | 0.436 | 0.446 | 0.413 | 0.434 | 0.495 | **-27.08%** |
>
> These results confirm that while capable tools are necessary, the **planning algorithm** (A* search with error recovery) is the decisive factor in handling complex workflows.

---

> > ### Author Response · Authors · 2025-11-27
> > **Follow up on the response**
> >
> > Dear Reviewer,
> >
> >
> > Thank you again for your time and helpful suggestions. We wanted to follow up to confirm that our responses have fully resolved your concerns. We remain available and are happy to provide any further clarifications should you have additional questions during the remaining discussion period.
> >
> >
> > Sincerely,
> >
> > The authors.

---

### Author Response · Authors · 2025-12-03
**General Response: Summary of Clarifications and Revisions**

Dear Reviewers and Area Chair,

We deeply appreciate the insightful and constructive comments provided by all reviewers.

We are encouraged by the reviewers' recognition of **CoSTA*** as a "**novel and promising**" approach (Reviewer `zfMS`) with a "**conceptually elegant**" hierarchical design (Reviewer `o7Vm`). Reviewers highlighted the "**practical scalability**" of our pruning mechanism (Reviewer `Gbp4`) and the clear motivation for balancing cost and quality in agentic workflows (Reviewer `BKHU`).

To address the shared concerns regarding the source of performance gains, technical novelty, and evaluation validity, we have provided detailed clarifications in our rebuttal. We have incorporated all clarifications into the revised manuscript, with updates highlighted in $\color{blue}{\text{blue}}$.

**Summary of Main Concerns of Reviewers and our Responses:**

*   **Do performance gains stem from tool diversity or planning? (Reviewers `o7Vm`, `zfMS`, `BKHU`): CoSTA*** **consistently outperforms baselines when constrained to the same toolsets as baselines.**
    We addressed this by detailing our **"Fair Comparison" Ablation (Appendix D, Tables 10-12)**. When CoSTA* is restricted to **exact the same toolset** adopted by the baselines, it still outperforms them by **~41%**. This empirically demonstrates that our hierarchical planning, specifically its ability to backtrack to previous subtasks and recover from errors is the critical differentiator over non-hierarchical, greedy ReAct-style agents.

*   **Scalability to Large Toolsets and Search Costs (Reviewers `o7Vm`, `Gbp4`): Our LLM-guided pruning reduces search complexity to $O(3^d)$, ensuring efficiency regardless of tool library size.**
    We clarified that our core novelty is the architectural solution to search intractability. Standard search is intractable ($O(|Tools|^d)$). CoSTA* utilizes LLM-guided pruning to reduce the branching factor to only relevant subtasks, lowering complexity to **$O(3^d)$**. This allows CoSTA* to scale to large tool libraries (1000+ tools) and deep task hierarchies (validated on 10-15 subtasks) without exploding computational cost.

*   **Are the results generalizable across different benchmarks? (Reviewers `o7Vm`, `BKHU`): Results hold across external benchmarks and rigorous human evaluation.**
    We verified that our results are not artifacts of a specific dataset by validating CoSTA* on standard external benchmarks (**MagicBrush** and **EMU-Edit**): CoSTA* outperforms the best baseline reported (e.g., **0.95 accuracy vs. 0.81** on EMU-Edit, see Appendix A). Furthermore, our internal evaluation utilizes a standardized human scoring rubric with low variance (**0.07**) across human raters, ensuring objective assessment.

*   **Reproducibility (Reviewer `o7Vm`):**
    We have committed to releasing the full codebase, the curated benchmark dataset, and the necessary data structures (Model Description Table and Benchmark Table) to ensure easy reproducibility of our results.


We believe these clarifications directly address the core concerns raised. We remain committed to incorporating these details into the final manuscript and welcome any further discussion.

Sincerely,

Authors.

---

### Meta-Review · Program_Chairs · 2026-01-08

**Summary:**

This submission presents CoSTA*, a hierarchical agent that combines LLM-based subtask decomposition with cost-sensitive A* search for multi-turn image editing, aiming to optimize quality–cost trade-offs. Reviewers acknowledge the clear motivation, solid engineering, and strong empirical results, as well as the authors’ thorough rebuttal with additional ablations clarifying that planning contributes substantially to the gains. However, multiple reviewers raise concerns that the technical novelty is incremental, with core components relying on well-established techniques, and that the benchmark and cost modeling may limit generality. In addition, an irregular and potentially hallucinated reference was flagged and requires correction (noted below). Overall recommendation: Reject.

**Reviewer Concerns:**

**Reviewer Concerns**

Concerns largely addressed by the rebuttal:

1. Several reviewers questioned whether performance gains stem from tool diversity rather than planning; the added fair-comparison ablations convincingly show that hierarchical planning and error recovery contribute substantially.
2. Concerns about scalability were partially addressed through clearer complexity analysis and explanations of LLM-guided pruning.
3. Reproducibility issues were mitigated by commitments to release code, data, and benchmarks.

Concerns that remain outstanding:

1. The core technical novelty is still viewed as incremental, as the method mainly integrates existing components without a fundamentally new algorithmic insight.
2. The motivation and definition of cost (primarily inference time) remain debatable for real-world image editing scenarios.
3. Questions about benchmark generality and potential tailoring to the proposed system are not fully resolved.

**Reviewer Scores:**

Reviewer o7Vm: Likely unchanged. The rebuttal addressed concerns about planning vs. tool diversity and evaluation clarity, but doubts about incremental novelty would likely remain.

Reviewer zfMS: Likely unchanged. The reviewer was already marginally positive, and while the clarifications on scalability and generalization help, they do not fundamentally alter the perceived contribution.

Reviewer Gbp4: Likely unchanged. The rebuttal adequately responded to scalability and runtime questions, but the core concern about limited algorithmic novelty would probably persist.

Reviewer BKHU: Likely unchanged. The authors’ defense of cost motivation and planning effectiveness is reasonable, but skepticism about problem framing and novelty would likely remain.

---

### Decision · Program_Chairs · 2026-01-26

Reject